# STOCHASTIC LAYER-WISE LEARNING: SCALABLE AND EFFICIENT ALTERNATIVE TO BACKPROPAGATION

## ABSTRACT

Backpropagation underpins modern deep learning, yet its reliance on global gradient synchronization limits scalability and incurs high memory costs. In contrast, fully local learning rules are more efficient but often struggle to maintain the cross-layer coordination needed for coherent global learning. Building on this tension, we introduce Stochastic Layer-wise Learning (SLL), a layer-wise training algorithm that decomposes the global objective into coordinated layer-local updates while preserving global representational coherence. The method is ELBO-inspired under a Markov assumption on the network, where the network-level objective decomposes into layer-wise terms and each layer optimizes a local objective via a deterministic encoder. The intractable KL in ELBO is replaced by a Bhattacharyya surrogate computed on auxiliary categorical posteriors obtained via fixed geometry-preserving random projections, with optional multiplicative dropout providing stochastic regularization. SLL optimizes locally, aligns globally, thereby eliminating cross-layer backpropagation. Experiments on MLPs, CNNs, and Vision Transformers from MNIST to ImageNet show that the approach surpasses recent local methods and matches global BP performance while memory usage invariant with depth. The results demonstrate a practical and principled path to modular and scalable local learning that couples purely local computation with globally coherent representations.

## 1 INTRODUCTION

The success of deep learning across a wide range of domains has been substantially driven by backpropagation (BP), a foundational learning algorithm enabling hierarchical representation learning through end-to-end gradient-based optimization Rumelhart et al. (1986); LeCun et al. (2015). Despite its algorithmic clarity and practical effectiveness, BP requires the exact storage of intermediate activations and subsequent gradient computation across all layers. This mechanism facilitates global credit assignment Lillicrap et al. (2020); it also introduces a well-known bottleneck called *update-locking* Jaderberg et al. (2017); Griewank & Walther (2008), where the weight update of a given layer must wait until both the forward pass through the entire network and the backward pass through deeper layers are complete. Consequently, this global dependency limits asynchronous updating, and imposes substantial memory and computational overhead, ultimately reducing training efficiency and scalability, especially in resource-constrained devices Luo et al. (2024); Belilovsky et al. (2019); Bengio et al. (2006).

BP is often seen as biologically implausible and this drives efforts to discover local learning rules for credit assignment in inspired by real neural systems Lillicrap et al. (2020); Scellier & Bengio (2017); Guerguiev et al. (2017). At the same time, neuroscience suggests that feedback connections may approximate global errors via local activity differences Guerguiev et al. (2017); Whittington & Bogacz (2019), hinting at a bioplausible path to deep learning Lillicrap et al. (2020); Sacramento et al. (2018). Yet, these approaches struggle to reconcile local updates with global learning and lack a unifying theoretical framework.

Given this context, a central research question emerges: *"Can we design a theoretical framework capable of decomposing deep neural network training into local (layer-wise) optimizations while retaining the benefits of hierarchical representation learning?"* This question captures a fundamental conflict: while local learning encourages architectural scalability and computational parallelization,

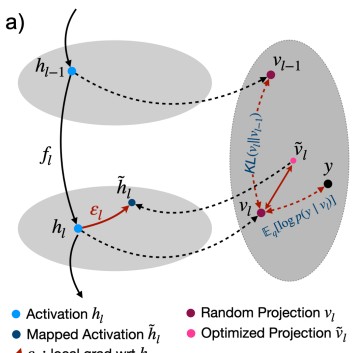 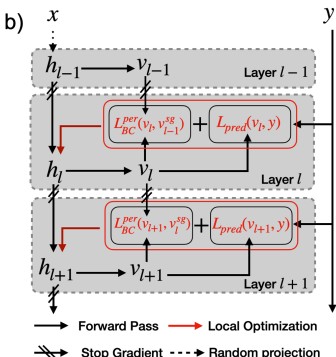

Figure 1: **Overview of Stochastic Layer-wise Learning (SLL).** (a) SLL treats each hidden activation $h_l$ as a latent variable and projects it to $v_l$ via a random matrix. The local ELBO comprises a log-likelihood term and a KL surrogate that promotes inter-layer consistency. Optimizing this loss yields the improved projection $\tilde{v}_l$ and its corresponding activation $\tilde{h}_l$. (b) SLL optimizes each layer independently using a prediction loss $\mathcal{L}_{\text{pred}}(v_l, y)$ from the log-likelihood and a feature alignment loss $\mathcal{L}_{\text{BC}}^{per}(v_l, v_{l-1}^{\text{sg}})$ approximating the KL term. Arrows denote forward computation (black), local updates (red), and stop-gradient paths (slashed).

effective deep learning relies on non-linear coordination across the entire network. This disconnect often results in misaligned learning signal and suboptimal performance, challenging the network scalability of locally trained models Yang et al. (2024).

To address this question, we ground local learning in how information propagates through deep networks: as signals traverse layers, raw inputs are progressively transformed into increasingly disentangled and class-separable representations He & Su (2023); Razdaibiedina et al. (2023); Telgarsky (2016). This refinement suggests that intermediate layers perform latent inference, selectively preserving task-relevant signals while suppressing redundancy Shwartz-Ziv & Tishby (2017). In this paper, we make this intuition precise by exhibiting a network-level ELBO that decomposes into layer-wise terms under a Markov assumption of network architecture, thereby furnishing principled local objectives while retaining an explicit link to the global goal. Building on this decomposition, we introduce **Stochastic Layer-wise Learning (SLL)**, a local learning framework in which each layer produces auxiliary categorical posteriors via fixed stochastic random projections, and the intractable layer-wise KL in the ELBO is replaced by a Bhattacharyya surrogate Bhattacharyya (1943) computed on these induced posteriors, yielding an ELBO-inspired and numerically stable update. Here, the projections preserve minibatch geometry with high probability by the Johnson–Lindenstrauss (JL) lemma Johnson et al. (1984); Razdaibiedina et al. (2023), which justifies computing divergences in the compressed space; we further apply multiplicative dropout to the fixed projection, which provides stochastic regularization consistent with the dropout-as-variational-inference interpretation Gal & Ghahramani (2016); we do not learn mask parameters and do not claim a variational bound over masks, and the overall objective remains ELBO-inspired at the layer level. SLL thus reconciles local optimization with hierarchical coordination, mitigating over-compression associated with direct KL minimization, maintaining global representational coherence, and enabling scalable, parallel training without full backpropagation.

This work targets mathematical analysis, algorithmic development, and experimental evaluations, leading to three principal contributions: **Theoretical contribution**: we formally decompose the network ELBO into layer-wise terms under a Markov assumption and prove that the arithmetic mean of these layer-wise ELBOs provides a valid lower bound on the global ELBO, establishing the theoretical basis for local training. **Algorithmic contribution**: we proposed SLL and demonstrate its potential as a scalable and efficient alternative to BP. By integrating stochastic random projections, SLL replaces the need for a complete backward pass, thereby facilitating structured local learning. **Experimental evaluations**: We demonstrate that SLL scales effectively across architectures and datasets, from MLPs on MNIST to ViTs on ImageNet. Our results show that the SLL algorithm surpasses recently proposed local training methods that address the update locking problem of BP. Moreover, SLL approaches or equals the accuracy performance of BP but with a significant reduction in memory (4× or more).

## 2 BACKGROUND

In supervised learning tasks, such as classification applications or regression, neural networks are designed to construct mappings between given input data $X$ and the corresponding target label $Y$. Traditional feedforward neural networks have a sequential structure in which each layer processes the output of the previous layer through a parameterized function. Following the classical formulation Rumelhart et al. (1986), such a $L$-layer neural network can be expressed as a chain of its parameterized sub-functions:

$$f_{1:L}(x) := f(f(...f(x, \theta_1)..., \theta_{L-1}), \theta_L) \tag{1}$$

where $\theta_i \in \Theta$ represents a set of learnable parameters at layer $i$. This hierarchical or Markov structure introduces a sequence of hidden representations $\mathcal{H} = [h_1, h_2, ..., h_L]$ where each representation is defined recursively as $h_i = f(h_{i-1}, \theta_i)$. Given the stacked structure of neural networks, each layer builds on the representation of the previous layer. This structure induces a hierarchical representation where higher layers encode increasingly abstract and task-relevant features.

**Backpropagation** is the standard approach for network training, aiming to optimize the parameters $\Theta$ of the network given a dataset of input-label pairs $(x, y)$ and a task-relevant loss function $\mathcal{L}(h_L, y)$. During training, input data is propagated through the entire network to generate predictions. The loss function then evaluates the network performance by quantifying the distance between these predictions and labels. Next, BP computes the gradient of the loss with respect to each parameter by recursively applying the chain rule in reverse through the network. The update rule for the parameters at layer $i$, $\theta_i$, are updated iteratively using gradient descent:

$$\theta'_i = \theta_i + \eta \Delta \theta_i; \quad \Delta \theta_i = \frac{\partial \mathcal{L}}{\partial \theta_i} = \frac{\partial \mathcal{L}}{\partial h_i} \cdot \frac{\partial h_i}{\partial \theta_i} = \frac{\partial \mathcal{L}}{\partial h_L} \prod_{j>i} \frac{\partial h_{j+1}}{\partial h_j} \cdot \frac{\partial h_i}{\partial \theta_i} \tag{2}$$

where $\eta$ is the learning rate. The first term (blue) captures the global contributions of activation $h_i$ to the global loss. It encodes dependencies across all subsequent layers and ensures that updates are coordinated with the global objective. The second term (red) reflects the local sensitivity of $h_i$ with respect to the corresponding parameters $\theta_i$, and can be calculated independently at each layer.

## 3 METHODOLOGY

In this section, we break the global training objective into local layer updates, so each layer learns locally while still contributing to the overall optimization of the network.

### 3.1 FROM GLOBAL LOSS TO GLOBAL ELBO

In principle, BP's inefficiencies arise from its treatment of activations as fixed, deterministic values that require explicit gradient computations across all layers. Here, we adopt a probabilistic formulation where each hidden activation is modeled as stochastic latent variables, conditioned on its previous layer. This hierarchy views forward computation as an approximate inference over latent variables, similar to the approaches in deep-generative models Kingma & Welling (2014); Sønderby et al. (2016). Thus, instead of optimizing deterministic activations, learning becomes an inference problem where the goal is to infer their posterior distributions conditioned on observed inputs and outputs. Formally, this corresponds to estimating the *true posterior* over the hidden representations:

$$p(h_1, \ldots, h_L \mid x, y) = \frac{p(y \mid h_L)p(h_L \mid h_{L-1}) \ldots p(h_1 \mid x)}{p(y \mid x)} = \prod_{i=1}^{L+1} p(h_i \mid h_{i-1})/p(y|x)$$

(Assumption 1)

where $h_0 := x$ and $h_{L+1} := y$. This joint distribution factorizes into a global *evidence* term and a product of local conditional terms. However, computing the evidence term requires marginalization over all hidden representations: $p(h \mid x, y) = \int \cdots \int \prod_{i=1}^{N+1} p(h_i \mid h_{i-1}) \, dh_L \ldots dh_1$ which is computationally intractable in high-dimensional deep architecture.

To address this challenge, we apply Variational Inference (VI) Blei et al. (2017); Ranganath et al. (2014) to approximate the intractable true posterior $p(y \mid x)$ with a variational surrogate distribution $q(h)$ by minimizing the KL divergence between them in latent space:

$$KL(q(h)\|p(h \mid x)) = \mathbb{E}_q[\log q(h)] - \mathbb{E}_q[\log p(h \mid x)].$$

where $\mathbb{E}_q[\cdot]$ denotes expectation under the variational posterior $q(h)$. This leads to maximizing the Evidence Lower Bound (ELBO):

$$\arg\max_\theta \mathcal{E} = \mathbb{E}_q[\log p(y \mid h)] - KL(q(h)\|p(h)) \tag{3}$$

where $p(h)$ is the prior distribution over latent variables. At this point, network optimization is reformulated as a structured variational inference problem, fundamentally distinct from standard BP.

## 3.2 FROM GLOBAL ELBO TO LAYER-WISE ELBO

**Generative and recognition models.** We view the network as a hierarchical latent variable model with generative transitions $p(h_i \mid h_{i-1})$ for $i = 1, \ldots, L$ and likelihood $p(y \mid h_L)$. To approximate the intractable posterior, we adopt a Markov assumption on the network architecture that mirrors the forward architecture Vahdat & Kautz (2020):

$$q(h_1, \ldots, h_L \mid x, y) = \prod_{i=1}^{L} q(h_i \mid h_{i-1}), \tag{Assumption 2}$$

where each factor may include auxiliary noise (reparameterization) or reduce to a delta, as specified below. Here $p(h_i \mid h_{i-1})$ denotes the generative transition (prior) at layer $i$, and $q(h_i \mid h_{i-1})$ is the approximate posterior (inference distribution) over $h_i$ given $h_{i-1}$. Under this factorization, a standard network-level variational objective is

$$\mathcal{E}_{NN} = \mathbb{E}_q\big[\log p(y \mid h_L)\big] - \sum_{i=1}^{L} \mathrm{KL}\big(q(h_i \mid h_{i-1}) \,\|\, p(h_i \mid h_{i-1})\big), \tag{Assumption 3}$$

with expectation over $q(h_1, \ldots, h_L \mid x, y)$. Each additive item admits a local interpretation, motivating the following layer-wise ELBO-inspired objective:

$$\mathcal{E}_i = \underbrace{\mathbb{E}_{q(h_i \mid x, y)}[\log p(y \mid h_i)]}_{\text{Expected log-likelihood}} - \underbrace{\mathrm{KL}\big(q(h_i \mid h_{i-1}) \,\|\, p(h_i \mid h_{i-1})\big)}_{\text{Layer-wise divergence}}, \tag{4}$$

where the first term encourages class-discriminative representations at layer $i$, and the second term regularizes by enforcing local consistency with $p(h_i \mid h_{i-1})$. In short, each layer learns to improve the prediction while remaining consistent with its generative prior Eldan & Shamir (2016).

**Layer–to–Network Relation.** Assume:

(A1) **Shared variational family.** All conditionals of $q$ are drawn from the same family across depths, and the same predictive head $p_\phi(y \mid \cdot)$ is used to evaluate all ELBO terms.

(A2) **Layer-/block-Markov posterior (analysis only).** For every $i$, $q(h_{i-1} \mid h_{i:L}, x) = q(h_{i-1} \mid h_i, x)$ (when using blocks, the equality is imposed at block boundaries).

(A3) **Monotone predictive gain.** For every $i$, $\mathbb{E}\big[\log p_\phi(y \mid h_i)\big] \geq \mathbb{E}\big[\log p_\phi(y \mid h_{i-1})\big]$, with expectations taken as in the ELBO definitions.

(A4) **KL budget.** The sum of the (conditional) KL terms used in the layerwise ELBOs upper-bounds the joint regularizer in the global ELBO: $\sum_i \mathbb{E}\big[\mathrm{KL}\big(q(h_{i-1} \mid h_i, x) \,\|\, p(h_{i-1} \mid h_i)\big)\big] \geq \mathbb{E}\big[\mathrm{KL}\big(q(h_{1:L} \mid x) \,\|\, p(h_{1:L})\big)\big]$.

Then the arithmetic mean of the layer-wise ELBOs is upper-bounded by the global ELBO: $\frac{1}{L}\sum_{i=1}^{L} \mathcal{E}_i \leq \mathcal{E}_{NN}$. Consequently, under (A1)–(A4), decreasing the local objectives provably tightens the global lower bound.

*Proof sketch.* By (A3), the average of the predictive terms (evaluated with the same head) is no larger than the top-level predictive term. By the chain rule of KL, (A2) and (A4) upper-bound the global KL by the sum of layer-wise conditional KLs. Combining yields arithmetic mean $\frac{1}{L}\sum_{i=1}^{L} \mathcal{E}_i \leq \mathcal{E}_{\mathrm{NN}}$. The inequality can fail when the layer-Markov assumption is violated, i.e., when $I_q(h_{i-1}; h_{i+1:L} \mid h_i, x) > 0$. This occurs in architectures with long-range cross-layer couplings such as hierarchical VAEs with top-down inference (e.g., Ladder-VAE Sønderby et al. (2016)), and encoder–decoder models with long lateral skip connections (U-Net Ronneberger et al. (2015)).

## 3.3 STOCHASTIC LAYER-WISE LEARNING (SLL)

To approximate the layer-wise ELBO in Assumption 3 with a strictly local training rule, we make each layer-wise KL term as a tractable surrogate defined on auxiliary discrete posteriors coming from adjacent layers. For layer $i$, we attach a random lightweight classification head $R_i : \mathbb{R}^{d_i} \to \mathbb{R}^K$ and define two categorical distributions over $K$ codes induced from the activations: a predictive prior $p_i(\cdot \mid h_{i-1}^{\text{sg}}) = \text{softmax}(R_{i-1} h_{i-1}^{\text{sg}})$ that depends only on the stop-gradient parent $h_{i-1}^{\text{sg}}$ (i.e. frozen input from the previous layer), and an auxiliary posterior $q_i(\cdot \mid h_i) = \text{softmax}(R_i h_i)$ which depends on the current activations $h_i$. We replace $\text{KL}\big(q(h_i \mid h_{i-1}) \| p(h_i \mid h_{i-1})\big)$, the KL term in the ELBO, by the per-sample Bhattacharyya surrogate:

$$\mathcal{L}_{\text{BC}}^{\text{per}}(i) = -\frac{1}{B} \sum_{b=1}^{B} \log \text{BC}\big(q_i^{(b)}, p_i^{(b)}\big), \qquad \text{BC}(u,v) = \sum_{k=1}^{K} \sqrt{u_k v_k} \in [0,1].$$

Here BC denotes the *Bhattacharyya coefficient*, introduced by Bhattacharyya Bhattacharyya (1943) as a measure of affinity between distributions; it equals the inner product of square-rooted probabilities. It is closely related to the squared Hellinger distance, since $H^2(u,v) = 1 - \text{BC}(u,v)$ Bhattacharyya (1943); van Erven & Harremoës (2014). This construction preserves locality because $p_i$ depends only on the frozen inputs $h_{i-1}^{\text{sg}}$, while also serving as a proxy for the ELBO term. A second-order expansion yields $\text{KL}(q\|p) = 4\big(1 - \text{BC}(q,p)\big) + o(\|q - p\|^2)$. Moreover, the inequalities $\text{KL}(q\|p) \geq -2 \log \text{BC}(q,p) \geq 2\big(1 - \text{BC}(q,p)\big)$ provide global monotone control and improved numerical stability, especially when probabilities are small. The resulting layer objective becomes:

$$\arg \min_{\theta} \mathcal{L}_i = \mathcal{L}_{\text{pred}} + \lambda_i \mathcal{L}_{\text{BC}}^{per} = \underbrace{\mathcal{L}_{\text{pred}}(R_i h_i, y)}_{\text{expected likelihood term}} + \underbrace{\lambda_i \mathcal{L}_{\text{BC}}^{\text{per}}(i)}_{\text{surrogate for KL}(q\|p)}, \qquad \lambda_i \geq 0, \quad (5)$$

,where $\lambda_i \geq 0$ is an *optional per-layer trade-off coefficient* that controls the *relative weight* of the alignment term (we use $\lambda_i = 1$ as default). This objective is ELBO-inspired rather than a strict ELBO lower bound. In general, optimizing $\{\mathcal{L}_i\}_{i=1}^{L}$ provides a structured approximation to the layer-wise ELBOs in Assumption 3 and, together with lemma 1, links these local updates to the global objective $\mathcal{E}_{NN}$, thereby enabling scalable training that remains faithful to the hierarchical variational formulation. Unlike auxiliary heads, greedy training, or reconstruction-based target propagation, our local objective is relational across depth which enforces *adjacent-layer probabilistic alignment* by minimizing a Bhattacharyya KL-surrogate between induced posteriors with stop–gradient on the parent, thereby regularizing inter-layer information flow via a proper, contractive $f$-divergence while preserving strict locality.

**Stochastic Random Projection.** We compute layer-wise divergences in a compressed subspace using fixed random projections, which preserve minibatch geometry with high probability by the JL lemma Johnson et al. (1984). Concretely, activations are mapped as $v_i = \frac{1}{\sqrt{d'}} R_i h_i$ with $R_i \in \mathbb{R}^{K \times d}$ sampled once at initialization with i.i.d. subgaussian entries, where $K \ll d$ and we set $K$ to the number of classes. For any finite set $\mathcal{H}$ of size $n$ (e.g., a minibatch), the JL lemma ensures that if $d' \geq C \varepsilon^{-2} \log(n/\delta)$ then, with probability at least $1 - \delta$, pairwise distances and inner products among $\{v_i(u) : u \in \mathcal{H}\}$ are preserved up to $O(\varepsilon)$; this justifies computing our alignment divergence on the auxiliary posteriors in the projected space. The projections act as lightweight heads that enable strictly local updates without backpropagating across layers. To improve generalization, we inject structured noise into the projection during training:

$$v_i = \frac{1}{\sqrt{d'}} (M_i \odot R_i) h_i, \qquad M_i \sim \text{Bernoulli}(p)^{d' \times d},$$

which acts as multiplicative dropout on the projection weights. This introduces Monte Carlo variability without learning the projection, and is consistent with the Bayesian view of dropout as approximate variational inference while our overall objective remains ELBO-inspired Gal & Ghahramani (2016). In our implementation it functions as a stochastic regularizer that stabilizes the induced posteriors and improves robustness. The result is a geometry-preserving, parameter-efficient mechanism that stabilizes alignment, mitigates over-compression, and scales local training.

**Implementation note.** Our implemented loss does not optimize $\mathcal{E}_j$ directly: it minimizes the (expected) *Rényi-$\frac{1}{2}$ / Bhattacharyya* divergence the student and stop-gradient parent posteriors in the shared $K$-class label space. With softmax and normalization, decreasing BC loss $\mathcal{L}_{BC}$ , and at

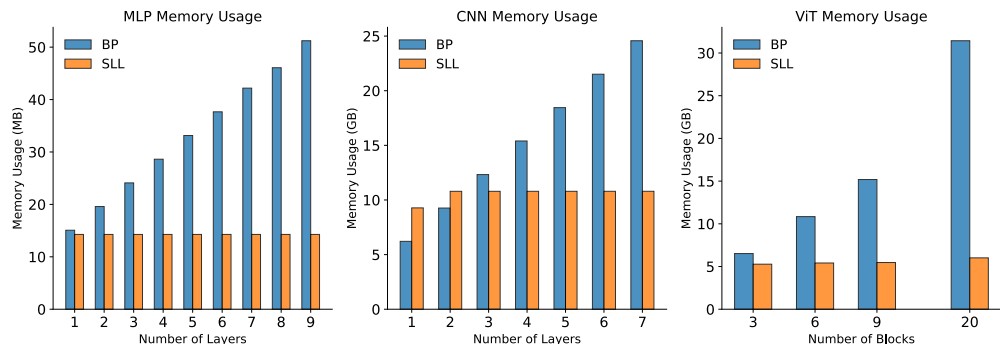

Figure 2: Peak training memory on (a) MLPs (1024 neurons/layer) as a function of depth. BP memory scales linearly, while SLL remains constant; (b) CNNs on the Imagenette without pooling layers. Each convolution layer uses a kernel size of 3 and 64 output channels; (c) ViTs on Imagenette. For fair comparison, we are using SGD as the optimizer in training.

the same time, pushing KL divergence towards zero in the label space; When (A1)–(A4) hold, these layer-wise alignments correlate with improvements in $\{\mathcal{E}_j\}$ and thereby tighten the global bound.

We use a deterministic approximate posterior $q(h_i \mid h_{i-1}) = \delta\big(h_i - f_i(h_{i-1})\big)$ and therefore compute the layer-wise divergence on the auxiliary categorical summaries $(q_i, p_i)$ rather than the continuous conditionals, preserving locality via stop–gradient on the prior side. During training, each layer is updated locally as the child-side distribution $q_i$, while its frozen output simultaneously serves as the parent-side target $p_{i+1}$ for the next layer, yielding a chain of coordinated adjacent-layer updates without cross-layer backpropagation.

## 4 RELATED WORK

The intersection of probabilistic inference and biologically plausible optimization has inspired a range of methods that seek to improve the scalability, interpretability, and local adaptability of deep learning. We organize related work into three areas: variational inference, local learning, and forward-only training. **Variational Inference and Probabilistic Deep Learning.** Variational inference (VI) enables tractable approximate Bayesian learning via ELBO maximization Blei et al. (2017); Jordan et al. (1999), foundational to deep generative models like VAEs Kingma & Welling (2014); Sohn et al. (2015); Higgins et al. (2017), and their structured extensions Sønderby et al. (2016); Vahdat & Kautz (2020). SLL approximates VI for feedforward networks, combining local latent approximations with task-driven learning, and can be seen as a layer-wise variational EM scheme. **Gradient-Based Local Learning** Local learning reduces backpropagation overhead by optimizing layers independently, from greedy layer-wise training Bengio et al. (2006) to local heads Belilovsky et al. (2019); Nøkland & Eidnes (2019) and synchronization strategies Ernoult et al. (2022). Recent blockwise and parallel approaches Yang et al. (2024); Apolinario et al. (2024) aim to scale under memory constraints, but often suffer from global feature inconsistency Yang et al. (2024). SLL alleviates this via variational alignment. More biologically inspired alternatives include FA Lillicrap et al. (2016), DFA Nøkland (2016), DPK Webster et al. (2021), and TP Lee et al. (2015), which replace gradients with alternative feedback signals. More recent Hebbian variants Journé (2023); Halvagal & Zenke (2023) show promise for scalable bio-plausible learning, though accuracy and depth remain challenges. **Forward-Only Credit Assignment** Forward-only methods eliminate backprop by using dual forward passes, e.g., Forward-Forward (FF)Hinton (2022), Signal PropagationKohan et al. (2023), and PEPITA (D&K'22). Other FF variants Wu et al. (2024); Dooms et al. (2023); Lee & Song (2023) reframe credit assignment via $L_2$ distances. Despite biological inspiration, these methods often face inter-layer misalignment Lorberbom et al. (2024), limiting hierarchical feature learning.

## 5 EXPERIMENTS

We evaluate the effectiveness, interpretability, and scalability of SLL across a range of standard benchmarks. Our experiments include multiple architectures, including MLPs, CNNs, and Vision

Transformers (ViTs), and datasets of increasing complexity, from MNIST LeCun et al. (1998) and CIFAR-10/100 Krizhevsky et al. (2009) to ImageNette and ImageNet-1K Deng et al. (2009). To assess SLL's capacity for local learning, we compare it against established local training baselines across multiple network scales. We further extend SLL to block-wise training (SLL+) for ViTs, demonstrating its compatibility with modern large-scale architectures without relying on full backpropagation.

| Method | Memory | FLOPS | MNIST | CIFAR10 | CIFAR100 |
|---|---|---|---|---|---|
| BP | $\mathcal{O}(NL)$ | $\mathcal{O}(N^2L)$ | $99.25 \pm 0.09$ | $60.95 \pm 0.33$ | $32.92 \pm 0.23$ |
| TP Lee et al. (2015) | $\mathcal{O}(NL)$ | $\mathcal{O}(N^2L)$ | $97.96 \pm 0.08$ | $49.64 \pm 0.26$ | - |
| FA Lillicrap et al. (2016) | $\mathcal{O}(NL)$ | $\mathcal{O}(NLC)$ | $98.36 \pm 0.03$ | $53.10 \pm 0.30$ | $25.70 \pm 0.20$ |
| DFA Nøkland (2016) | $\mathcal{O}(NL)$ | $\mathcal{O}(NLC)$ | $98.26 \pm 0.08$ | $57.10 \pm 0.20$ | $26.90 \pm 0.10$ |
| PEPITA(D&K'22) | $\mathcal{O}(NL)$ | $\mathcal{O}(N^2L)$ | $98.01 \pm 0.09$ | $52.57 \pm 0.36$ | $24.91 \pm 0.22$ |
| SF Kohan et al. (2023) | $\mathcal{O}(N)$ | $\mathcal{O}(N^2L)$ | $98.29 \pm 0.03$ | $57.38 \pm 0.16$ | $29.70 \pm 0.19$ |
| SLL | $\mathcal{O}(N)$ | $\mathcal{O}(NLC)$ | $\mathbf{99.32 \pm 0.05}$ | $\mathbf{61.43 \pm 0.31}$ | $\mathbf{32.95 \pm 0.26}$ |

Table 1: Performance and computational complexity of SLL vs prior local-learning methods for MLPs on MNIST, CIFAR-10, and CIFAR-100 under the same experimental setup. BP and baseline results are taken from (Kohan et al., 2023). Memory and FLOPs (credit-assignment compute only) are reported as asymptotic scaling in $N$ (neurons per layer), $L$ (layers), and $C$ (classes). Metrics are mean $\pm$ std over three runs. "–" denotes values not reported.

## 5.1 EXPERIMENTS ON MLPs

We begin by evaluating SLL on fully connected networks trained on benchmarks: MNIST and CIFAR-10/100. These datasets serve as controlled settings to study local learning dynamics in low-dimensional and moderately complex inputs.

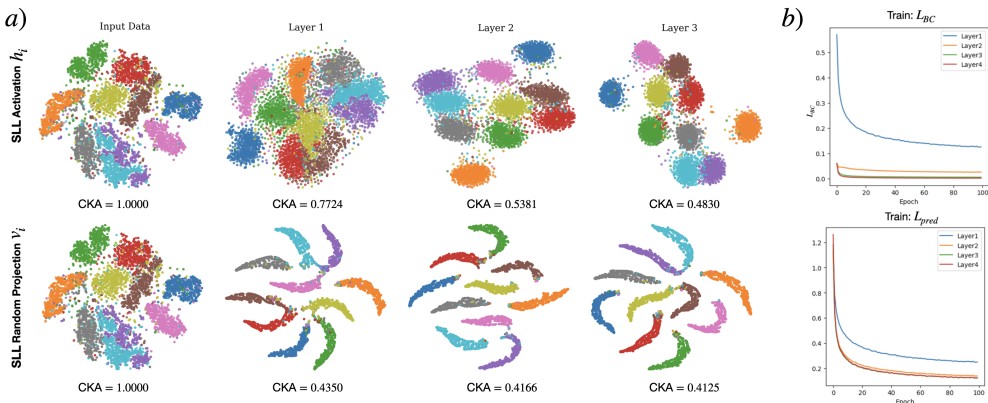

Figure 3: **Layer-wise evolution under SLL.** (a) t-SNE of a held-out batch: activations $h_i$ (top) and projected codes $v_i = R_i h_i$ (bottom). Class clusters tighten with depth and the readout preserves the class geometry. CKA is computed w.r.t. the input; higher = more similar to pixel space, lower = more abstract. (b) Per-layer training curves: both $\mathcal{L}_{\text{BC}}$ and $\mathcal{L}_{\text{pred}}$ decrease steadily and are lowest for deeper layers, indicating progressive adjacent-layer alignment without full backpropagation. More details check Fig.A7 in appendix.

**Accuracy and Efficiency.** To establish a comprehensive comparison, we evaluate SLL alongside a range of biologically motivated and local learning algorithms that do not rely fully or avoid BP. All models are trained with identical architectures and training schedules to ensure a fair comparison. As shown in Table 1 and Figure 4(a), SLL consistently outperforms all local learning baselines, despite operating under reduced memory and computational budgets. In particular, under this identical setting in Kohan et al. (2023), SLL even surpasses BP on these datasets while requiring fewer operations and avoiding global gradient synchronization. Moreover, Figure 2(a) confirms SLL's memory efficiency

during training. The training memory usage of SLL remains effectively constant as the depth of the network increases, in contrast to its theoretical complexity reported in Table 1.

**Representation Visualization.** We analyze the internal representations of the network trained by SLL in Figure 3(a). In general, input features are initially entangled, deeper layers show improved class separation. It is obvious that $v_i$ forms sharper, more distinct clusters than $h_i$, indicating that random projections not only preserve but often enhance class-discriminative structure.

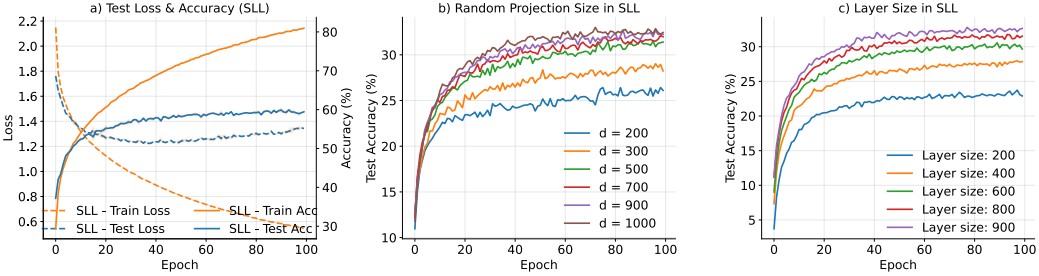

Figure 4: (a) Training curves of a 3-layer MLP on CIFAR-10 via SLL. Ablation study: (b) Effect of the *pooled feature size* $d$ in a $3 \times 1000$ MLP on CIFAR-100: before the head, each layer's activation is reduced to $d$ features by *adaptive average pooling*, then mapped by a fixed linear readout $R \in \mathbb{R}^{d \times K}$ (with $K$ the number of classes). (c) network width in SLL on CIFAR-100, showing that wider layers significantly enhance performance and stability.

**Ablation study.** We further investigate the effect of projection dimension and network width on SLL performance (Figure 4b,c). Increasing the projection dimension $d$ improves test accuracy, with diminishing returns beyond $d = 700$, suggesting a trade-off between representational precision and efficiency. Likewise, wider networks result in faster convergence and higher accuracy on CIFAR-100, with improvements saturating above 800 neurons. These trends are consistent with our theoretical insights in JL Lemma Johnson et al. (1984), which indicate that high-dimensional layers reduce alignment loss and preserve inter-layer information. Together, these findings highlight the role of capacity and compression in enabling stable local learning with SLL.

| Model | F-MNIST | CIFAR10 | CIFAR100 | Imagenette | Tiny-Imagenet$_{64}$ |
|---|---|---|---|---|---|
| BP-CNN | 93.52(0.22) | 91.58(0.53) | 68.7(0.38) | 90.5(0.45) | 48.15(0.82) |
| **Local Learning** | | | | | |
| FA (Nok'16) | 91.12(0.39) | 60.45(1.13) | 19.49(0.97) | – | – |
| DFA (Nok'16) | 91.54(0.14) | 62.70(0.36) | 48.03(0.61) | – | 32.12(0.66) |
| DKP (Web'21) | 91.66(0.27) | 64.69(0.72) | 52.62(0.48) | – | 35.37(1.92) |
| Softhebb (Jour'23) | – | 80.3 | 56 | 81.0 | – |
| SGR (Yan'24) | – | 72.40(0.75) | 49.41(0.44) | – | – |
| LLS (Apo'24) | 90.54(0.23) | 88.64(0.12) | 58.84(0.33) | – | 35.99(0.38) |
| **Forward-Only** | | | | | |
| FF-CNN (Hin'22) | – | 59 | – | – | – |
| TFF (Doo'23) | 91.44(0.49) | 83.51(0.78) | 35.26(0.23) | – | – |
| PEPITA (D&K'22) | – | 56.33(1.35) | 27.56(0.60) | – | – |
| LC-FF (Lor'24) | 88.4 | 48.4 | – | – | – |
| DF-R (Wu'24) | 92.5 | 84.75 | 48.16 | 81.2 | – |
| **SLL-CNN** | **93.67(0.17)** | **91.36(0.32)** | **67.57(0.18)** | **88.09(0.73)** | **49.42(0.65)** |

Table 2: CNN test accuracies comparing SLL with prior local-learning and forward-only methods. Values are reported as mean(std) over three runs; "–" indicates not reported.

## 5.2 SCALING SLL TO CNNS

We next explore how SLL can scale effectively to convolutional architectures despite discarding explicit spatial structure when utilizing fully connected random projection. To this end, we evaluate SLL on a VGG-11 architecture and compare it against representative local learning methods, forward-only training algorithms, and conventional global BP.

**Accuracy.** Table 2 reports the test accuracies in F-MNIST, CIFAR-10/100 and Tiny-Imagenet. SLL performs competitively with BP, achieving within 1–2% of BP on all datasets like F-MNIST, CIFAR-10/100 and TinyImageNet200, even slightly surpassing it on F-MNIST. In particular, SLL outperforms all local and forward-only baselines on all given tasks, including DFA Nøkland (2016), DKP Webster et al. (2021), SoftHebb Journé (2023), and TFF Dooms et al. (2023).

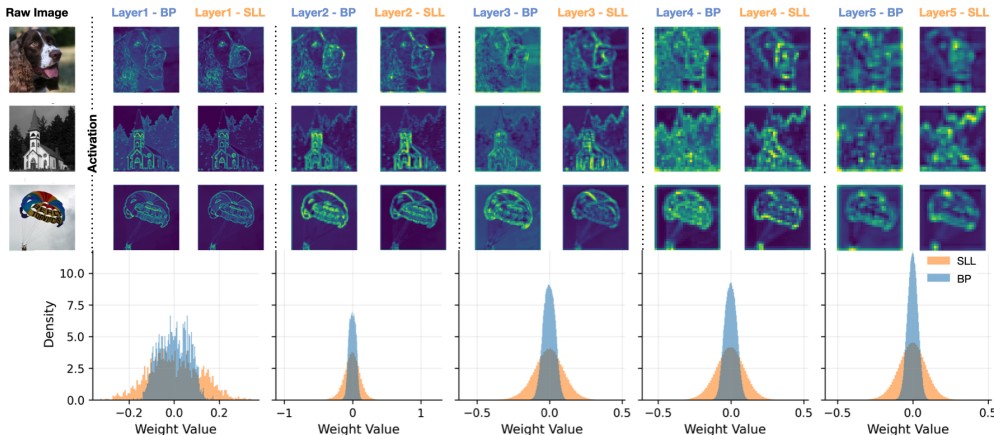

Figure 5: Activation and weight distributions from VGG-11 trained with BP and SLL on Imagenette.

**Training memory efficiency.** Figure 2(b) illustrates the training memory usage of SLL and BP on CNNs. While SLL exhibits a clear memory advantage in MLPs, its benefit is more moderate in CNNs. This is because convolution operations are inherently sparse and memory-efficient, while the dense random projections used in SLL introduce additional overhead. However, SLL still maintains a significant advantage in deeper architectures.

**Feature visualization.** Figure 5 indicates that SLL effectively learns high-quality spatial and discriminative representations, despite discarding explicit spatial priors. Compared with BP, the broader weight distributions from SLL suggest robust and distributed encoding.

## 5.3 SCALING TO VISION TRANSFORMER

Moreover, we use Vision Transformers (ViTs) Dosovitskiy et al. (2021) as a scalability benchmark for SLL, since their dense, MLP-like blocks and large activations footprints heavily impact compute and memory, making them ideal for testing efficiency and convergence.

To scale to ViTs, we propose $SLL^{i+}$, a blockwise variant of SLL tailored for large residual architectures. We partition the ViT architecture into $i$-units, each comprising one or more attention blocks; training is hybrid where standard backpropagation is used within each unit, while between units we optimize the local objectives independently, eliminating global backpropagation across the entire model. It effectively turns SLL into a local block-wise training scheme for deep networks, in this case ViTs. This design aligns with the residual structure of ViT while preserving the localized memory and learning advantage of SLL.

| Task | Method | Test Acc | Mem(GB) |
|------|--------|----------|---------|
| CIFAR-10 | BP | 93.62 | 3.05 |
| | $SLL^{7+}$ | 92.17 | **1.18(↓ 64.1%)** |
| CIFAR-100 | BP | 75.24 | 3.05 |
| | $SLL^{7+}$ | 74.27 | **1.18(↓ 64.1%)** |
| Imagenette | BP | 92.82 | 22.12 |
| | $SLL^{7+}$ | 92.25 | **5.43(↓ 75.45%)** |
| Imagenet | BP | 79.4 | 20.70 |
| | $SGR^{3+}$ | 78.65 | 11.73(↓ 43.33%) |
| | $SLL^{3+}$ | 72.43 | **6.54(↓ 68.41%)** |
| | $SLL^{12+}$ | 59.62 | **4.30(↓ 79.22%)** |

Table 3: ViTs results. "Mem" denotes peak GPU training memory. SGR refers to Yang et al. (2024). BP baseline of ImageNet is from Yuan et al. (2021).

$SLL^{i+}$ leverages the class token or mean over all tokens as a stable and semantically meaningful signal for local supervision. This allows efficient classification without requiring end-to-end backpropagation. As shown in table 3, $SLL^{i+}$ achieves large memory savings in Vision Transformers while preserving accuracy, with memory use staying nearly constant as block depth increases (Figure 2(c)). This trend is similar to the MLP findings and demonstrates SLL scalability across architectures. Compared to BP, $SLL^{i+}$ reduces training memory by 64%–80% without sacrificing stability or model capacity.

## 6 DISCUSSION AND CONCLUSIONS

The above results highlight open opportunities for improving SLL. First, the Markov assumption between layers, while simplifying inference, may limit expressivity in architectures with long-range dependencies such as residual connections like UNets Ronneberger et al. (2015). Second, the absence of second-order gradient information may reduce SLL's effectiveness in navigating ill-conditioned loss surfaces. Third, SLL's reliance on local supervision may limit convergence in large-scale classification tasks where informative gradients may only emerge in later layers. In addition, it is worth investigating hierarchic and epoch-dependent schedules for alignment weight $\lambda_i$ to improve optimization dynamics. Finally, aggressive dimension reduction via random projection may lead to information loss in narrow architectures. Addressing these challenges through more expressive dependency modeling, adaptive projection schemes, architecture-aware supervision, and specialized training approaches for sequential models could extend the applicability of SLL to broader research.

It is worth mentioning that the SLL also draws conceptual parallels with Equilibrium Propagation (EP) Scellier & Bengio (2017) and energy-based models. Both frameworks enable local updates that align with global objectives, but they operate through distinct mechanisms: stochastic layerwise updates for SLL and dynamical relaxation for EP. Bridging these perspectives under a unified probabilistic or dynamical systems framework is an interesting direction for future research.

In conclusion, we introduce SLL, a scalable and memory-efficient alternative to BP that reformulates training as an ELBO inspired, stochastic layer-wise learning. By combining stochastic random projection with a Bhattacharyya surrogate for the layer-wise KL, SLL enables parallel, local updates while preserving global coherence without global BP and without additional trainable parameters. Compared to BP, SLL achieves competitive accuracy with significant memory efficiency, up to 4× in our settings, and consistently outperforms prior local learning methods. It generalizes effectively across MLPs, CNNs, and ViTs, scaling from small to moderately large vision tasks. Beyond training efficiency, SLL also provides a structured probabilistic view of deep representations, offering a foundation for interpretable learning dynamics and architecture design grounded in information flow.

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
