# A APPENDIX

**Appendix Contents**

## A.1 ALGORITHM AND CODE

### A.1.1 PSEUDO CODE

---

**Algorithm 1** Stochastic Layer-wise Learning

---

**Require:** Training batch data $(x, y)$, learning rate $\eta$, random projection matrix $R_l \sim \mathcal{N}$
**Ensure:** Updated network weights $\theta$
1: **for** each layer $l$ from 1 to $L - 1$ **do**
2:     **Detach From above layer:** $h_{l-1} = h_{l-1}.detach()$
3:     **Update activation:** $h_l \leftarrow f(h_{l-1}, \theta_l)$
4:     **Approximate** $\partial h_l$:
5:         Random Projection: $v_l = dp(R_l)h_l$ or $v_l = dp(R_l)[h_l, y]$
6:         Loss: $\mathcal{L}_l \leftarrow L_{Pred}(v_l, y) + L_{BC}(\text{softmax}(v_l), \text{softmax}(v_{l-1}))$
7:         Activation drift: $\varepsilon_l \leftarrow \frac{\partial \mathcal{L}_l}{\partial h_l}$
8:     **Weight Update:** $\theta_l \leftarrow \theta_l - \eta \cdot \frac{\partial h_l}{\partial \theta_l} \varepsilon_l$
9:     Clear unnecessary tensors
10: **end for**
11: $h_{L-1} = h_{L-1}.detach()$
12: $h_L \leftarrow f(h_{L-1}, \theta_L)$
13: Loss: $\mathcal{L}_L \leftarrow L_{Pred}(h_L, y) + L_{fa}(h_L, v_{L-1})$
14: $\theta_L \leftarrow \theta_L - \eta \cdot \frac{\partial \mathcal{L}_L}{\partial \theta_L}$

---

Activation drift is The first term (blue) captures the global contributions of activation $h_i$ to the global loss in Eq:2.

### A.1.2 PYTHON CODE FOR BHATTACHARYYA COEFFICIENT

Listing 1: Loss function

```python
def L_BC_per(q: torch.Tensor, p: torch.Tensor,
             reduction: str = "mean",
             eps: float = 1e-12,
             detach_p: bool = True) -> torch.Tensor:
    """
    q, p: [B, K] probabilities (nonnegative; rows ~ sum to 1).
    """
    if detach_p:
        p = p.detach()
    q = (q.clamp_min(eps) / q.clamp_min(eps).sum(dim=-1, keepdim=True))
    p = (p.clamp_min(eps) / p.clamp_min(eps).sum(dim=-1, keepdim=True))
    # log BC via log-sum-exp for stability
    log_bc = torch.logsumexp(0.5 * (q.log() + p.log()), dim=-1)
    loss_per = -log_bc
    return (loss_per.mean() if reduction == "mean"
            else loss_per.sum() if reduction == "sum"
            else loss_per)
```

## A.2 ADDITIONAL EXPERIMENTAL RESULTS

### A.2.1 SMALL CNN

| Method | Model | CIFAR-10 | CIFAR-100 |
|---|---|---|---|
| BP | CNN Apolinario et al. (2024) | $87.57 \pm 0.13$ | $62.25 \pm 0.29$ |
| DFA | CNN Apolinario et al. (2024) | $71.53 \pm 0.38$ | $44.93 \pm 0.52$ |
| PEPITA | CNN Dellaferrera & Kreiman (2022) | $56.33 \pm 1.35$ | $27.56 \pm 0.60$ |
| LLS | CNN Apolinario et al. (2024) | $84.10 \pm 0.27$ | $55.32 \pm 0.38$ |
| SVP | CNN | $87.48 \pm 0.32$ | $59.74 \pm 0.27$ |

Table A4: Comparison of different methods on CIFAR-10 and CIFAR-100 using a 3-layer CNN. All methods are evaluated under the same network structure for a fair comparison.

### A.2.2 RESNET 50

| Method | CIFAR-10 |
|---|---|
| FA Kappel et al. | 70.3 |
| pred-Sim Kappel et al. | 92.4 |
| BLL Kappel et al. | 92.6 |
| BWBPF Cheng et al. (2024) | 95.15 |
| InfoProp Wang (2021) | 93.07 |
| AugLocal Ma et al. | 93.47 |
| SGR Yang et al. (2024) | 85.65 |
| **SLL$^{i+}$** | **95.46** |

Table A5: Blockwise training of **ResNet-50** on **CIFAR-10**: comparison with prior baselines. Values are Top-1 accuracy (%). downsampled ImageNet-1k (32×32) with a blockwise-trained ResNet, obtaining 53.14% top-1, 75.21% top-5.

### A.2.3 ABLATION STUDY ON CIFAR-100

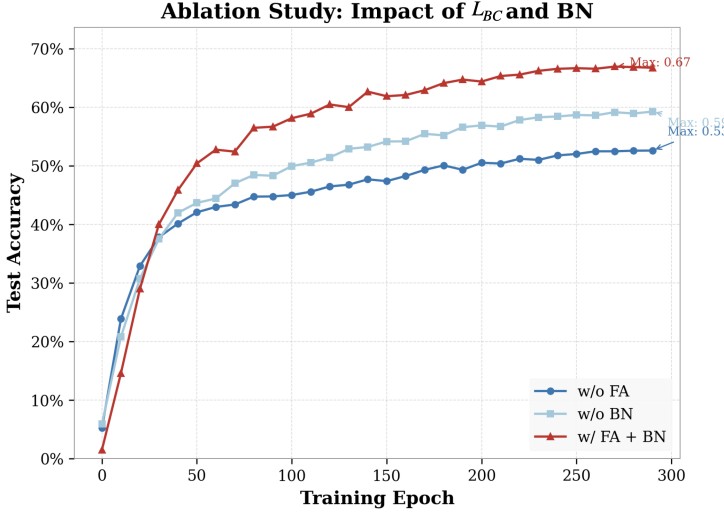

Figure A6: Ablation on CIFAR-100 showing that combining feature alignment loss ($\mathcal{L}_{BC}$) and batch normalization significantly improves test accuracy and convergence over variants without $\mathcal{L}_{BC}$ or BN.

### A.2.4 LAYER-WISE LOSS AND ACCURACY

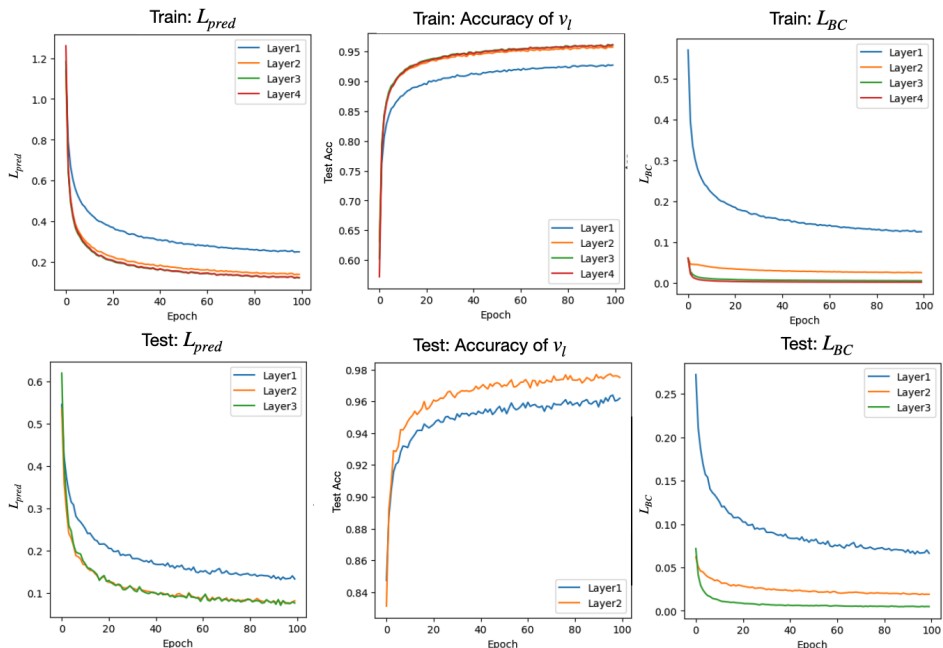

Figure A7: **Layer-wise training dynamics under SLL on MLP(MNIST).** Top: Learning curves for training dataset; bottom: earning curves for test dataset. Left: prediction loss $\mathcal{L}_{\mathrm{pred}}$ on projected codes $v_\ell = R_\ell h_\ell$. Middle: classification accuracy from the head on $v_\ell$. Right: Bhattacharyya alignment loss $\mathcal{L}_{\mathrm{BC}}$ between induced posteriors $(q_\ell, p_\ell)$. Curves are shown per layer; deeper layers achieve lower $\mathcal{L}_{\mathrm{pred}}$ and $\mathcal{L}_{\mathrm{BC}}$ and higher accuracy, indicating progressive local learning and strengthened inter-layer consistency without cross-layer backpropagation.

### A.3 ABLATION ON THE BC LOSS

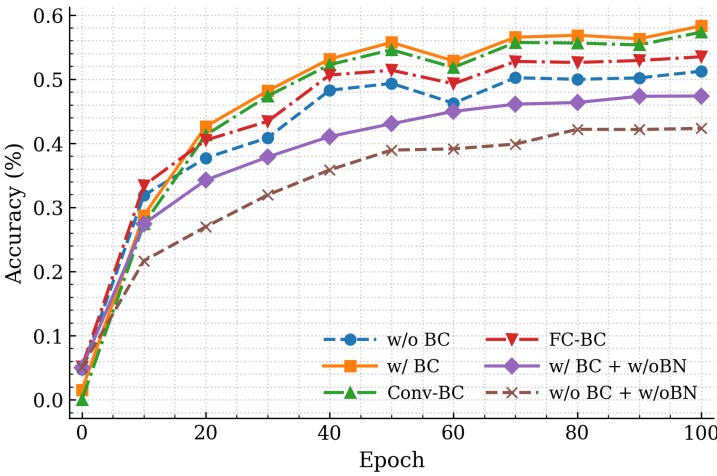

Figure A8: **Ablations of the Bhattacharyya alignment loss $\mathcal{L}_{\mathrm{BC}}$.** Top-1 accuracy versus epoch for the VGG model. *w/o BC*: baseline. *w/ BC*: $\mathcal{L}_{\mathrm{BC}}$ at all layers. *Conv-BC / FC-BC*: $\mathcal{L}_{\mathrm{BC}}$ applied only to convolutional or only to fully connected layers. *w/ BC + w/o BN* and *w/o BC + w/o BN*: BatchNorm removed to test interaction with $\mathcal{L}_{\mathrm{BC}}$. Adding $\mathcal{L}_{\mathrm{BC}}$ consistently accelerates learning and improves final accuracy; applying it on convolutional blocks (*Conv-BC*) yields the strongest gains, while removing BatchNorm degrades performance that is partially recovered by $\mathcal{L}_{\mathrm{BC}}$.

### A.3.1  ABLATION STUDY ON DROPOUT RATE

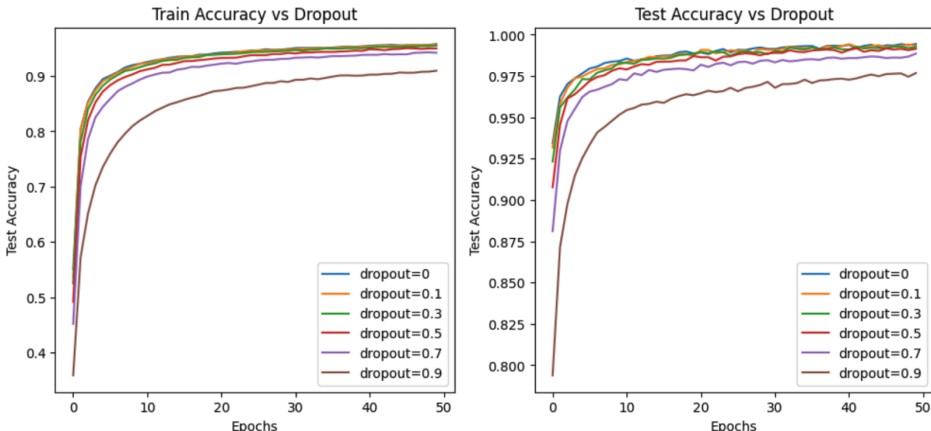

Figure A9: Dropout ablation for the SLL readout on MNIST with 3-layer MLP. Training (left) and test (right) accuracy over epochs for dropout probability $p \in \{0, 0.1, 0.3, 0.5, 0.7, 0.9\}$ applied to the projection head. SLL is robust for $p \leq 0.3$; $p = 0.5$ slightly slows early learning but reaches a similar final accuracy; aggressive dropout ($p \geq 0.7$) reduces signal-to-noise, slowing convergence and lowering the final accuracy, with $p = 0.9$ clearly harmful. We use $p = 0.1$ as the default in all main experiments.

### A.3.2  ASSUMPTION CHECK IN BP

These BP trends match the SLL dynamics in App. Fig. A7, providing empirical support that the assumption approximately holds in the forward networks.

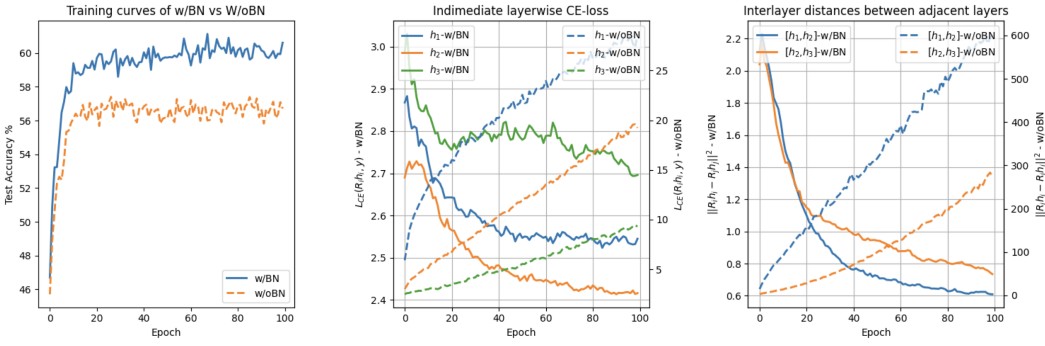

Figure A10: **BP check for the monotone predictive-gain assumption on an MLP (CIFAR-10).** *Left:* Test accuracy with BatchNorm (solid) vs. without BN (dashed). *Middle:* Per-layer predictive loss $\mathcal{L}_{\text{pred}}(R_i h_i, y)$ for $i \in \{1, 2, 3\}$ under the different fixed random readout $R_i \in \mathbb{R}^{d \times K}$; BN yields steady decreases across depths and epochs, while removing BN breaks monotonicity. *Right:* Inter-layer distances $D_i = \|R_i h_i - R_{i-1} h_{i-1}\|_2^2$ averaged over the batch; BN drives $D_i$ down (adjacent-layer alignment), whereas without BN they grow. These BP trends match the SLL dynamics in App. Fig. A7, supporting the assumption approximately holds in the MLPs.

## A.4 PROOF OF THEOREMS

### A.4.1 THEOREM 1: LAYER-WISE ELBO PROVIDES A VALID VARIATIONAL BOUND

*Let $\mathcal{L}_{NN}$ be the global Evidence Lower Bound (ELBO) of the network:*

$$\mathcal{E}_{NN} = \mathbb{E}_q[\log p(y \mid h_L)] - D_{\text{KL}}(q(\mathcal{H})\|p(\mathcal{H})). \tag{6}$$

*Then, the sum of layer-wise ELBOs in SVP provides a valid lower bound:*

$$\frac{1}{L}\sum_{i=1}^{L}\mathcal{E}_i \le \mathcal{E}_{NN}. \tag{7}$$

*Proof.* We start with the marginal likelihood:

$$\log p(y|x) = \log \int p(y, h_1, h_2, ..., h_L|x)\, dh_1 \ldots dh_L. \tag{8}$$

Here, we define $\mathcal{H} = \mathcal{H}_L = \{h_1, h_2, ..., h_L\}$ as the set of activations. Introducing the variational approximation $q(\mathcal{H}|x, y)$, we have:

$$\log p(y|x) = \log \int \frac{p(y, \mathcal{H}|x)}{q(\mathcal{H}|x, y)} q(\mathcal{H}|x, y)\, dh_1 \ldots dh_L. \tag{9}$$

Applying Jensen's inequality (since logarithm is a concave function):

$$\log p(y|x) \ge \int \log \frac{p(y, \mathcal{H}|x)}{q(\mathcal{H}|x, y)} q(\mathcal{H}|x, y)\, dh_1 \ldots dh_L = \mathbb{E}_{q(\mathcal{H}|x,y)}\left[\log \frac{p(y, \mathcal{H}|x)}{q(\mathcal{H}|x, y)}\right]. \tag{10}$$

Using the joint probability factorization:

$$p(y, \mathcal{H}|x) = \prod_{i=1}^{L+1} p(h_i|h_{i-1}, x), \tag{11}$$

where $h_0 = x$ and $h_{L+1} = y$ by convention.

Thus, we obtain the global ELBO:

$$\mathcal{E}_{NN} = \mathbb{E}_{q(\mathcal{H}|x,y)}\left[\log p(y|h_L, x) + \sum_{i=1}^{L}\log p(h_i|h_{i-1}, x) - \log q(\mathcal{H}|x, y)\right]. \tag{12}$$

Using the variational factorization assumption in SLL:

$$q(\mathcal{H}|x, y) = \prod_{i=1}^{L} q(h_i|h_{i-1}, x, y), \tag{Assumption 2}$$

where again $h_0 = x$.

From the *Assumption 1,2*, the global ELBO can be rewritten as:

$$\mathcal{E}_{NN} = \mathbb{E}_{q(\mathcal{H}|x,y)}\left[\log p(y|h_L, x) + \sum_{i=1}^{L}\log p(h_i|h_{i-1}, x) - \sum_{i=1}^{L}\log q(h_i|h_{i-1}, x, y)\right] \tag{13}$$

$$= \mathbb{E}_{q(\mathcal{H}|x,y)}\left[\log p(y|h_L, x)\right] + \sum_{i=1}^{L}\mathbb{E}_{q(\mathcal{H}|x,y)}\left[\log p(h_i|h_{i-1}, x) - \log q(h_i|h_{i-1}, x, y)\right]. \tag{14}$$

We can rewrite this in terms of KL divergence:

$$\mathcal{E}_{NN} = \mathbb{E}_{q(\mathcal{H}|x,y)}\left[\log p(y|h_L, x)\right] - \sum_{i=1}^{L}\mathbb{E}_{q(\mathcal{H}|x,y)}\left[\log q(h_i|h_{i-1}, x, y) - \log p(h_i|h_{i-1}, x)\right]. \tag{15}$$

where the expectation of the KL divergence terms can be rewritten as:

$$\mathbb{E}_{q(\mathcal{H}|x,y)}\left[\log q(h_i|h_{i-1},x,y) - \log p(h_i|h_{i-1},x)\right] \tag{16}$$

$$=\mathbb{E}_{q(\mathcal{H}_i|x,y)}\left[D_{\text{KL}}(q(h_i|h_{i-1},x,y)\|p(h_i|h_{i-1},x))\right]. \tag{17}$$

Therefore, the global ELBO becomes:

$$\mathcal{E}_{NN} = \mathbb{E}_{q(\mathcal{H}|x,y)}\left[\log p(y|h_L,x)\right] - \sum_{i=1}^{L}\mathbb{E}_{q(\mathcal{H}_i|x,y)}\left[D_{\text{KL}}(q(h_i|h_{i-1},x,y)\|p(h_i|h_{i-1},x))\right]. \tag{18}$$

Now, let's define the layer-wise ELBO for each layer $i$:

$$\mathcal{E}_i = \mathbb{E}_{q(\mathcal{H}_i|x,y)}\left[\log p(y|h_i,x)\right] - \mathbb{E}_{q(\mathcal{H}_i|x,y)}\left[D_{\text{KL}}(q(h_i|h_{i-1},x,y)\|p(h_i|h_{i-1},x))\right]. \tag{19}$$

Summing the layer-wise ELBOs:

$$\sum_{i=1}^{L}\mathcal{E}_i = \sum_{i=1}^{L}\mathbb{E}_{q(\mathcal{H}_i|x,y)}\left[\log p(y|h_i,x)\right] - \sum_{i=1}^{L}\mathbb{E}_{q(\mathcal{H}_i|x,y)}\left[D_{\text{KL}}(q(h_i|h_{i-1},x,y)\|p(h_i|h_{i-1},x))\right]. \tag{20}$$

**Assumption 3: Monotone predictive gain under consistent measure.** This assumption is well-founded in established theoretical results showing that neural network expressivity grows exponentially with depth, with deeper representations providing exponentially more representational capacity than shallow ones Telgarsky (2016); Eldan & Shamir (2016). Then we assume: For all $i < L$:

$$\mathbb{E}_{q(h_i|x,y)}[\log p(y|h_i,x)] \le \mathbb{E}_{q(h_L|x,y)}[\log p(y|h_L,x)] \tag{Assumption 3}$$

**Assumption 4: KL budget constraint.** This constraint ensures that the accumulated KL regularization cost across all layers does not exceed the total predictive improvement gained from using the full network depth, preventing the variational bound from becoming arbitrarily loose due to excessive regularization. So, we assume:

$$\frac{L-1}{L}\sum_{i=1}^{L}\mathbb{E}_{q(h_{i-1}|x,y)}\left[D_{\text{KL}}\big(q(h_i|h_{i-1},x,y)\|p(h_i|h_{i-1},x)\big)\right] \tag{Assumption 4}$$

$$\le \frac{1}{L}\sum_{i=1}^{L}\big(\mathbb{E}_{q(h_L|x,y)}[\log p(y|h_L,x)] - \mathbb{E}_{q(h_i|x,y)}[\log p(y|h_i,x)]\big)$$

**Proof.** **Step 1**: Using the Markov factorization, the global ELBO becomes:

$$\mathcal{E}_{NN} = \mathbb{E}_{q(h_L|x,y)}[\log p(y|h_L,x)]$$
$$- \sum_{i=1}^{L}\mathbb{E}_{q(h_{i-1}|x,y)}\left[D_{\text{KL}}\big(q(h_i|h_{i-1},x,y)\|p(h_i|h_{i-1},x)\big)\right] \tag{21}$$

where $q(h_L|x,y)$ is the marginal of the joint distribution $q(\mathcal{H}|x,y)$ under the Markov factorization.

**Step 2**: Define

$$A_i := \mathbb{E}_{q(h_i|x,y)}[\log p(y|h_i,x)] \tag{22}$$

$$K_i := \mathbb{E}_{q(h_{i-1}|x,y)}\left[D_{\text{KL}}\big(q(h_i|h_{i-1},x,y)\|p(h_i|h_{i-1},x)\big)\right] \ge 0 \tag{23}$$

Then:

$$\mathcal{E}_i = A_i - K_i \tag{24}$$

$$\mathcal{E}_{NN} = A_L - \sum_{i=1}^{L}K_i \tag{25}$$

**Step 3**: Compute the difference:

$$\frac{1}{L}\sum_{i=1}^{L}\mathcal{E}_i - \mathcal{E}_{NN} = \frac{1}{L}\sum_{i=1}^{L}(A_i - K_i) - \left(A_L - \sum_{i=1}^{L}K_i\right) \tag{26}$$

$$= \frac{1}{L}\sum_{i=1}^{L}A_i - A_L - \frac{1}{L}\sum_{i=1}^{L}K_i + \sum_{i=1}^{L}K_i \tag{27}$$

$$= \frac{1}{L}\sum_{i=1}^{L}A_i - A_L + \frac{L-1}{L}\sum_{i=1}^{L}K_i \tag{28}$$

$$= -\frac{1}{L}\sum_{i=1}^{L}(A_L - A_i) + \frac{L-1}{L}\sum_{i=1}^{L}K_i \tag{29}$$

**Step 4**: Apply the assumptions:

- By **Assumption 3**: Each $A_L - A_i \geq 0$, so $-\frac{1}{L}\sum_{i=1}^{L}(A_L - A_i) \leq 0$

- By **Assumption 4**: $\frac{L-1}{L}\sum_{i=1}^{L}K_i \leq \frac{1}{L}\sum_{i=1}^{L}(A_L - A_i)$. Justification: On average across layers, the extra coding cost—quantified by the KL between $q$ and $p$, i.e., $\frac{L-1}{L}\sum_{i=1}^{L}K_i$—of shaping the latents must be no larger than the predictive gain—the increase in expected log-likelihood, i.e., $\frac{1}{L}\sum_{i=1}^{L}(A_L - A_i)$. In short, depth's benefit pays for its inference complexity.

**Step 5**: Conclude:

$$\frac{1}{L}\sum_{i=1}^{L}\mathcal{E}_i - \mathcal{E}_{NN} = -\frac{1}{L}\sum_{i=1}^{L}(A_L - A_i) + \frac{L-1}{L}\sum_{i=1}^{L}K_i \leq 0 \tag{30}$$

Therefore: $\frac{1}{L}\sum_{i=1}^{L}\mathcal{E}_i \leq \mathcal{E}_{NN}$

This completes the proof that the mean of layer-wise ELBOs provides a valid lower bound for the global ELBO.

## A.5 MODEL ARCHITECTURE AND EXPERIMENT DETAILS

### A.5.1 COMPUTER RESOURCES

All experiments were conducted on a single NVIDIA A100 GPU with 40GB of memory. No multi-GPU or distributed training was used.

### A.5.2 DATASETS

In this paper, we evaluate SVP on a range of benchmark datasets.

- MNIST is a handwritten digit image dataset over 10 classes including 60,000 images for training and 10,000 images for testing. Each image is a $28 \times 28$ gray-scale image.

- Fashion MNIST contains fashion items images such as clothing and shoes. It consists of a training set of 60,000 grayscale images and a test set of 10,000 images. Each image has a $28 \times 28$ size and is categorized into 10 classes.

- CIFAR-10 consists of $32 \times 32$ RGB images for object recognition with 50,000 images for training and 10,000 images for testing. It has 10 classes, with 5,000 training and 1,000 testing images per class.

- CIFAR-100 comprises a total of 60,000 $32 \times 32$ RGB images distributed across 100 classes. Within each class, 500 images are allocated for training, while 100 images are for testing.

- Tiny-ImageNet is a downsampled subset of ImageNet to a size of $64 \times 64$. This dataset consists of 200 classes and each class contains 500 images for training and 100 images for testing.

- ImageNet-1Kis a large-scale image classification dataset containing over 1.28 million training images and 50,000 validation images across 1,000 classes. Images are typically resized to 224×224 pixels for training.

### A.5.3 MLP

**Architecture** For MNIST, we use a 2-layer MLP with 800 neurons per layer and ReLU activations, followed by dropout. For CIFAR-10 and CIFAR-100, we adopt a 3-layer MLP with 1000 neurons per layer, also using ReLU activations and dropout regularization.

**Experimental Details.** During training, we apply random horizontal flipping and normalization as standard data augmentation. Models are optimized using the Adamax optimizer with a learning rate of 0.001, trained for 100 epochs across all datasets.

### A.5.4 CNN

The architecture and training configurations used for CNN experiments are summarized in Table A6.

| Dataset | Network Architecture | Training Details |
|---|---|---|
| FMNIST
SIZE: 28×28
CLASS: 10 | Conv64k3 - MaxPool2 - Conv128k3 - MaxPool2 - Conv256k3 - Conv256k3 - MaxPool2 - Conv512k3 - MaxPool2 - FC10 | Data Aug: Normalize
Optimizer: Adam
Learning Rate: 0.003
Batch Size: 128
Epochs: 100 |
| CIFAR10/100
SIZE: 32×32
CLASS: 100 | Conv256k3 - MaxPool2 - Conv512k3 - MaxPool2 - Conv512k3 - Conv1024k3 - MaxPool2 - Conv1024k3 - MaxPool2 - Conv1024k3 - Conv1024k3 - FC2048 - FC100 | Data Aug: Crop, Flip, AutoAugment, Normalize
Optimizer: Adam
Learning Rate: 0.003
Batch Size: 128
Epochs: 500 |
| Imagenette
SIZE: 224×224
CLASS: 10 | Conv64k3 - MaxPool2 - Conv128k3 - MaxPool2 - Conv256k3 - Conv256k3 - MaxPool2 - Conv512k3 - MaxPool2 - Conv512k3 - Conv1024k3 - FC10 | Data Aug: Crop, Flip, Normalize
Optimizer: Adam
Learning Rate: 0.001
Batch Size: 128
Epochs: 500 |
| Tiny-Imagenet
SIZE: 64×64
CLASS: 200 | Conv64k3 - MaxPool2 - Conv128k3 - MaxPool2 - Conv256k3 - Conv256k3 - MaxPool2 - Conv512k3 - MaxPool2 - Conv512k3 - Conv1024k3 - FC4096 - FC200 | Data Aug: Crop, Flip, Normalize
Optimizer: Adam
Learning Rate: 0.001
Batch Size: 128
Epochs: 500 |

Table A6: Network architectures and training configurations for CNNs.

### A.5.5 VIT

Details of the ViT architecture and experimental setup are provided in Table A7.

| CIFAR10/100
SIZE: 32×32
CLASS: 100 | blocks=7,
heads=12,
mlp-ratio=2,
embedding=384 | Data Aug: Crop, Flip, Au-
toAugment, Normalize
Optimizer: AdamW
Learning Rate: 0.003
Batch Size: 128
Epochs: 500 |
|---|---|---|
| Imagenette
SIZE: 224×224
CLASS: 10 | blocks=7,
heads=12,
mlp-ratio=2,
embedding=384 | Data Aug: Crop, Flip, Nor-
malize
Optimizer: AdamW
Learning Rate: 0.001
Batch Size: 128
Epochs: 500 |
| Imagenet
SIZE: 224×224
CLASS: 1000 | blocks=12,
heads=6,
mlp-ratio=3,
embedding=384 | Data Aug: Crop, Flip, Nor-
malize
Optimizer: AdamW
Learning Rate: 0.0005
Batch Size: 256
Epochs: 500 |

Table A7: Network architectures and training configurations for ViTs.