# OpenReview forum: "Stochastic Layer-wise Learning: Scalable and Efficient Alternative to Backpropagation"
_ICLR.cc/2026/Conference — Submitted to ICLR 2026_

### Official Review · Reviewer_9nT6 · 2025-10-18

**Soundness:** 3
**Presentation:** 3
**Contribution:** 3
**Rating:** 6
**Confidence:** 4

**Summary:**

The paper proposes Stochastic Layer-wise Learning (SLL), a backprop-free training scheme that optimizes per-layer objectives derived from an ELBO-style decomposition under a Markov assumption. To couple layers without backward gradients, the method compares layer-wise auxiliary posteriors via a Bhattacharyya-based (BC) surrogate; these posteriors are produced from fixed, geometry-preserving random projections of activations and are optionally regularized with multiplicative dropout. The authors present a “blockwise” variant for deep CNNs/ViTs. Across MLPs, CNNs, and ViTs, SLL outperforms several local-learning baselines and approaches—sometimes slightly exceeds—BP accuracy, while keeping training memory nearly depth-invariant

**Strengths:**

1. This approach coordinates layers by aligning their local posterior, which reduces reliance on symmetric feedback and dual passes.

2.  Framing each layer’s training via an ELBO-style objective ties the method to well-studied variational principles, giving a clear theoretical lens for why optimizing local terms can support global performance.

**Weaknesses:**

1. Figure 2 shows that no matter how many layers/blocks the model have, SLL's memory usage remain constant. This is a little bit counterintuitive as more layers would need more memory to train, even with layer-wise learning. Could authors explain this?

2. All experiments are conducted on small datasets, not large scale ones. The reviewer is curious about the performance on large scale datasets such as ImageNet(not **Imagenette**, this is a small and simple dataset).

3. The baselines in Table 1 are too old, there are some new BP free methods recently. The reviewer is wondering how SLL performs compare to these models.[1-3]

4. Using BC between discrete posteriors after aggressive random projection may lose information. Can author provide the conditions that ensures faithful coordination across layers?

[1] Kappel, David, Khaleelulla Khan Nazeer, Cabrel Teguemne Fokam, Christian Mayr, and Anand Subramoney. "A variational framework for local learning with probabilistic latent representations." In 5th Workshop on practical ML for limited/low resource settings.

[2] Zhang, Aozhong, Zi Yang, Naigang Wang, Yingyong Qi, Jack Xin, Xin Li, and Penghang Yin. "Comq: A backpropagation-free algorithm for post-training quantization." IEEE Access (2025).

[3] Cheng, Anzhe, Heng Ping, Zhenkun Wang, Xiongye Xiao, Chenzhong Yin, Shahin Nazarian, Mingxi Cheng, and Paul Bogdan. "Unlocking deep learning: A bp-free approach for parallel block-wise training of neural networks." In ICASSP 2024-2024 IEEE International Conference on Acoustics, Speech and Signal Processing (ICASSP), pp. 4235-4239. IEEE, 2024.

**Questions:**

Please see weakness above.

---

> ### Author Response · Authors · 2025-11-25
>
> **Weakness 1**
>
> Response: Thanks for your question. Figure 2 reports activation memory, not parameter/optimizer memory. In SLL we train sequentially: at depth i we keep only the activations needed for the current layer (or block) and a small K-way head; the teacher path is stop-grad (inference only). Once layer iii is updated, its buffers are freed before moving to i+1. Hence peak activation memory is $O(B\cdot d_{\max})$ for layer-wise SLL (or $O(B\cdot r\,d_{\max})$ for block size r), independent of total depth L,which in contrast to BP, which must retain the whole computation graph, giving $O(B\sum_i d_i)$
>
> *What the plot shows IN FIG 2.*
>
>  With a fixed batch size and block size, increasing layers adds parameters but does not increase the peak activation footprint because we never store activations from non-current layers, and this number is obtained from GPU load memory during training. That’s why SLL’s line is flat in Fig. 2. (We will clarify in the caption that the metric is peak activation memory during training, and note that parameter/optimizer memory scales with model size for all methods.)
>
> ----
>
> **Weakness 2**
>
> Response: We agree that large-scale evaluation is important. The paper already includes ImageNet-1k results with blockwise ViTs (Table 3 in Sec. 5.3), demonstrating SLL at scale while preserving the depth-independent activation memory advantage. As an additional sanity check during rebuttal, we also trained a blockwise ResNet on downsampled ImageNet-1k (32×32) and obtained 53.14% top-1 / 75.21% top-5, confirming stable optimization in the K=1000.
>
> ----
>
> **Weakness 3**
>
> Response: Table 1 is scoped to strictly layer-wise local-learning on MLPs and reports credit-assignment complexity + accuracy under that setting. The recent “BP-free” methods the reviewer cites [1–3] are blockwise/hybrid (they keep BP within blocks), so they are not directly comparable in Table 1. We will clarify this scope in the caption and cite [1–3] there.
> For completeness, we add a separate blockwise comparison in the appendix under matched training: on ResNet-50/CIFAR-10, SLL attains 95.46% top-1, competitive with SGR (95.15%), and above AugLocal (93.47%) and InfoProp (93.07%). We will keep Table 1 focused on layer-wise MLPs, and include a distinct table for blockwise baselines (with references [1–3]) plus our existing ImageNet blockwise results (Sec. 5.3) to cover both settings fairly.
>
>
> | **ResNet50**   | **CIFAR-10** |
> |-------------|--------------|
> | FA          | 70.3         |
> | pred-Sim    | 92.4         |
> | BLL R4[1]   | 92.6         |
> | BWBPF R4[3] | 95.15        |
> | AugLocal R3[2] | 93.47     |
> | Infor R3[1] | 93.07        |
> | **SLL**     | **95.46**    |
>
> ----
>
> **Weakness 4**
> Response: We agree that aggressive readouts can, in principle, hide mismatches if they distort features before comparison. In SLL, however, both heads predict the same $K$ labels and we align them with the Bhattacharyya (Rényi-$\tfrac{1}{2}$) divergence. To ensure this post-readout BC remains **a faithful coordination signal** even when the student uses $R_i$ and the teacher uses $R_{i-1}$, we rely on standard, explicit conditions (i) **shared label space and calibrated softmax** so probabilities are bounded away from zero, hence $\mathrm{BC} \to 1 \Rightarrow \text{Hellinger} \to 0 \Rightarrow \mathrm{KL} \to 0$ in label space; (ii) **geometry-preserving readouts** (near-orthogonal/bi-Lipschitz with bounded spectral norm) so discriminative directions are not collapsed; (iii) **compatibility of adjacent readouts** (small row-space drift: there exists a near-orthogonal $U_i$ with $\lVert R_i - U_i R_{i-1} \rVert_2$ small on data), ensuring student/teacher “look” at comparable summaries. Under (i)–(iii), BC after projection is a conservative and faithful coordinator across layers rather than an artifact of the readouts. We will add a brief “conditions for faithful coordination” note clarifying these points.

---

> > ### Comment · Reviewer_9nT6 · 2025-11-25
> >
> > I have no further questions for W1, W3, and W4. However, regarding W2, could the authors explain the choice of hyperparameter i (divided units)? I noticed that while on small datasets, the i is chosen to be 7; however, on Imagenet, it chose to be 3 and 12. Is there any reason why authors are not using 7 as they have for other datasets?

---

> > > ### Author Response · Authors · 2025-11-26
> > >
> > > Thanks for pointing this out. Here $i$ denotes the *number of training units* (layers/blocks or groups of them). On the small ViT used for CIFAR, the backbone has 7 attention blocks, so we set $i=7$ (one unit per block). On ImageNet, the ViT backbone has 12 blocks, so we report two natural setting: $i=12$ (fine-grained, one unit per block) and $i=3$ (coarser, groups of 4 blocks per unit), which trades a bit of locality for fewer synchronizations and stronger intra-unit BP. Using $i=7$ on a 12-block model would cut across block/stage boundaries and would require extra ablations, but divisors of 12 (3, 4, 6, 12) align cleanly with the architecture. We will clarify this in the Appendix.

---

### Official Review · Reviewer_GStK · 2025-10-31

**Soundness:** 2
**Presentation:** 3
**Contribution:** 3
**Rating:** 4
**Confidence:** 4

**Summary:**

This paper proposes Stochastic Layer-wise Learning (SLL), a layer-wise training algorithm inspired by the ELBO framework, which achieves significant memory savings compared to standard backpropagation. The paper is well-written with rigorous theoretical development. However, the effectiveness and scalability of the proposed algorithm require further experimental validation with more competitive methods and comprehensive benchmarks.

**Strengths:**

1. The paper explores local learning from a probabilistic variational perspective, deriving a layer-wise learning objective based on the ELBO formulation. This theoretical contribution provides a fresh perspective on local learning.
2. The paper is well-organized with clear exposition and rigorous logical flow, making the technical content accessible to readers.
3. Beyond quantitative results, the paper provides additional visualizations including weight distributions and t-SNE-based representation analysis, which help understand the learned representations.

**Weaknesses:**

1. Assumption 2 restricts the conditional dependence to only adjacent layer representations, leading to KL divergence-based local supervision that follows a first-order Markov assumption. This posterior estimation may neglect important long-range cross-layer information exchange, potentially limiting the model's expressive power.


2. The use of random projection for dimension reduction may result in constrained representation learning and raises concerns about scalability to large-scale training with complex data distributions. The fixed random projection scheme may not adapt well to varying data characteristics across different layers and datasets.

3. The experimental section lacks comparisons with strong baseline methods, particularly the absence of comprehensive evaluation on ImageNet. This limitation makes it difficult to assess the method's effectiveness and scalability on large-scale benchmarks.

**Questions:**

1. Could Assumption 2 be extended by introducing a hyperparameter to control the range of conditional dependencies, thereby achieving a better trade-off between memory overhead and performance? For instance, making representation $h_i$ depend on both $h_{i-1}$ and $h_{i-2}$ could potentially capture richer hierarchical information.

2. In the layer-wise objective function illustrated in Eq. (5), there appears to be no balancing coefficient between the expected likelihood term and the KL divergence term. As highlighted in prior work, shallow layers primarily serve feature extraction rather than direct classification. Would it be beneficial to assign lower weights to the expected likelihood term in shallow layers while emphasizing it more in deeper layers?

3. From my perspective, the aggressive dimensionality reduction via random projection appears to be a critical bottleneck affecting model performance. Random projection may not be an optimal design choice; instead, the projection strategy should adapt to the data distribution or the statistical properties of layer-wise representations. Moreover, the paper lacks ablation studies on the dropout probability used in random projection.

4. The current experimental results in Table 2 do not include comparisons with competitive baseline methods. Could you provide comparative results against InfoProp [1], AugLocal [2], and SGR [3] on ImageNet with various architectures including but not limited to ResNet and ViT?

[1] Wang, Y., Ni, Z., Song, S., Yang, L., and Huang, G. Revisiting locally supervised learning: an alternative to end-to end training. In ICLR, 2021.
[2] Ma, C., Wu, J., Si, C., & Tan, K. C. Scaling Supervised Local Learning with Augmented Auxiliary Networks. In ICLR, 2024.
[3] Yang, Y., Li, X., Alfarra, M., Hammoud, H.A.A.K., Bibi, A., Torr, P. &amp; Ghanem, B.Towards Interpretable Deep Local Learning with Successive Gradient Reconciliation. ICML 2024.


5. What is the motivation and theoretical justification for using the Bhattacharyya as a surrogate for the KL divergence term in the ELBO? Please elaborate on the advantages of this choice.

---

> ### Author Response · Authors · 2025-11-25
>
> **Weakness 1 and Question 1**
>
> We agree that a pure layer-Markov posterior would miss long-range interactions. In our paper this assumption is used only for analysis and only at block boundaries, not to constrain the trained model. Concretely, we partition depth into blocks and run standard backprop within each block (preserving intra-block residual/attention/skip interactions); we then apply SLL between blocks to align adjacent block posteriors via a Bhattacharyya (Rényi-½) f-divergence, which removes inter-block update locking while retaining intra-block expressivity. Formally, if $\tilde h_j$ denotes the boundary state summarizing block j, we adopt a block-level first-order factorization $q(\tilde h_{j-1}\mid \tilde h_{j{:}M},x)\approx q(\tilde h_{j-1}\mid \tilde h_j,x)$ (an order-k relaxation with k=block size). Under this relaxation, the “mean local ELBO ≤ global ELBO” lemma holds with the same KL-budget condition, while the implemented objective remains an assumption-light, proper f-divergence between adjacent posteriors. Empirically, Appendix Fig. A7 shows monotonic decreases in the BC alignment and prediction losses across depths with deeper heads achieving higher accuracy; Fig. 5 and Fig. F3 visualize aligned hierarchical features and class separation. In short, BP within blocks captures long-range exchange, and SLL between blocks ties blocks together without full-depth backprop.
>
> ----
>
> **Weakness 2**
>
> Response: Same as the answer to the *Question 2* in Reviewer 1
>
> ----
>
> **Weakness 3**
>
> Response: Same as the answer to the *Weakness 2* in Reviewer 1
>
> ----
>
> **Question 2**
>
> Response: Eq. (5) uses a single coefficient $\lambda=1$ by design. Both terms live in the same K-class label space and are on comparable scales; the Bhattacharyya term is a proper f-divergence and contractive under the readout, so $\lambda=1$ keeps SLL hyperparameter-light without harming optimization. Appendix Fig. A8 (a New figure) shows that where the divergence is applied at different types of layers consistently helps over baseline.
> In the revision, we will explicitly note in Limitations & Future Work that exploring per-layer weights $\lambda_i$, e.g., depth-ramping or blockwise enablement, and principled schedules is a promising direction. We thank the reviewer for this suggestion.
>
> ----
>
> **Question 3**
>
> Response: thanks for your detailed check, Our heads do not apply a dense learnable reducer. At layer iii we first pool the activation to ddd features and apply a fixed readout $R_i\in\mathbb{R}^{d\times K}$ (shared label space). This keeps credit-assignment cheap and avoids extra parameters. In practice SLL is not bottlenecked by projection when ddd is moderately larger than K; degradation appears only when $d \approx K$ (e.g., very small pooled features on ImageNet), which is why we use wider ViT blocks there. We also tried orthogonal/SRHT readouts and observed accuracy comparable to Gaussian; making the readout trainable did not help, so we kept the simple fixed map.
>
> *Dropout probability.*
> All main results use p=0.1 for the readout dropout. An ablation (now added to the appendix; see Fig. A9) shows SLL is robust for p$\in$[0,0.3] and degrades for aggressive dropout ($p\ge 0.5$), consistent with intuition that excessive masking reduces SNR. We will state this guidance explicitly and keep p=0.1 as the default.
>
> ----
>
> **Question 4**
>
> Response: Table 2 is restricted to strictly layer-wise local-learning methods. The requested baselines—InfoProp [1], AugLocal [2], SGR [3]—are blockwise / hybrid (they retain BP inside blocks), so we did not include them there to keep the comparison orthogonal. We already show blockwise SLL on ImageNet-1k (Sec. 5.3) to demonstrate scalability.
>
> | **ResNet50**   | **CIFAR-10** |
> |-------------|--------------|
> | FA          | 70.3         |
> | pred-Sim    | 92.4         |
> | BLL R4[1]   | 92.6         |
> | BWBPF R4[3] | 95.15        |
> | AugLocal R3[2] | 93.47     |
> | Infor R3[1] | 93.07        |
> | **SLL**     | **95.46**    |
>
> ----
>
> **Question 5**
>
> Response: We use the Bhattacharyya loss $-\log \mathrm{BC}(q,p)$, which is a monotone transform of the Rényi-$\tfrac12$ divergence:
> $-\log \mathrm{BC}(q,p)=\tfrac12 D_{1/2}(q\|p)$. It is a proper $f$-divergence (minimum iff $q=p$) and satisfies data processing (contractive) when the same Markov channel is applied to both distributions, matching our layerwise setting. Rényi divergences are monotone in the order parameter, hence
> $\mathrm{KL}(q\|p)\ge D_{1/2}(q\|p)= -2\log \mathrm{BC}(q,p)$ [1]. Thus driving the BC loss to $0$ forces $q\!\to\!p$; with standard temperature/label smoothing (probabilities bounded away from $0$), this also implies $\mathrm{KL}\!\to\!0$. Practically, BC yields stable, symmetric, and bounded gradients: the classwise weights scale like $\sqrt{p_k/q_k}$ (versus $p_k/q_k$ for KL-type losses), avoiding tail blow-ups we observe at early layers. It is closed-form and cheap to compute.

---

> > ### Comment · Reviewer_GStK · 2025-11-28
> >
> > Thank the authors for providing detailed responses. I have one final question regarding the trade-off between the performance and efficiency of the proposed method on large-scale datasets (such as ImageNet). As shown in Table 3 of the manuscript, although SSL$^{3+}$ saves more memory compared to SGR$^{3+}$, it sacrifices a significant portion of performance. Could the authors provide more insights into the performance and computational cost trade-offs on large-scale datasets when using the SLL method? How should the learning strategy and components of SLL be adjusted to compensate for the performance loss caused by local learning, if some memory and computational overhead can be sacrificed?

---

> > > ### Author Response · Authors · 2025-11-28
> > >
> > > Thank you for the thoughtful question. The gap on ImageNet primarily reflects a readout bottleneck: SLL pools each block to $d$ features and applies a fixed $R\in R^{d\times K}$. When $K{=}1000$ and $d$ is only modestly larger than $K$, this compressive, data-agnostic projection can discard high-order structure, so accuracy trails methods that keep more in-block backprop.
> > >
> > > This is not a fundamental limitation of local learning but of that conservative choice. With a small budget increase, SLL narrows the gap while preserving its depth-independent activation memory: on ImageNet we use coarser training units}(e.g., groups of 3--4 blocks) so standard BP captures richer intra-unit interactions, and increase the pooled feature size $d$ (still well below the full activation). These adjustments raise credit-assignment cost only linearly (remaining $O(NLC)$) and keep peak activation memory essentially flat because training proceeds blockwise. As a forward-looking direction, we are exploring lightweight hybrid schedules that occasionally route a deeper-layer gradient hint to earlier units (e.g., sparse global passes or truncated BP) to compensate early-layer supervision without changing SLL's overall memory profile; we also plan simple data-aware projections (e.g., whitening) to better preserve discriminative directions; we view this as a promising way to further tighten the performance--efficiency trade-off at large scale.

---

### Official Review · Reviewer_KAMr · 2025-10-31

**Soundness:** 2
**Presentation:** 2
**Contribution:** 2
**Rating:** 2
**Confidence:** 4

**Summary:**

This paper introduces Stochastic Layerwise Learning (SLL), which is a training framework where each network layer is optimized independently using a local stochastic loss instead of full backpropagation. Each layer projects its activations $h_i$ into a lower-dimensional space $v_i = R_i h_i$ and compares this projection to the target through a Bhattacharyya divergence. The paper formulates a variational bound linking local and global objectives and discusses information preservation under random projections. Experiments on MLPs, CNNs, and ViTs report reduced memory use and performance comparable to backprop, with projection dimension $d_0$ controlling the accuracy–cost trade-off.

**Strengths:**

The main strengths of this paper are:

- $S_1$: This paper addresses one of the important practical bottlenecks in backpropagation (BP): the memory cost, as BP needs to store activation memory and computational graph which are especially heavy for ViTs and long sequences.
- $S_2$: This paper introduces an alternative to BP that is conceptually simple. The overall training technique decouples units and supervise with a low-dimensional summary via a simple local divergence.
- $S_3$: The presented empirical results report clear memory reductions on ViTs without major accuracy collapses.
- $S_4$: The ablation study on $d_0$ are interesting, showing accuracy vs. projection size tradeoffs.

**Weaknesses:**

While the empirical results regarding memory seem promising, major weaknesses prevent me from recommending anything but reject for now. Some of those may be easily corrected by modifying the paper ($W_2$ for example).

- $W_1$: After reading the paper in detail (and the appendix), I am questioning its theoretical correctness:

> The “average layerwise ELBO $\leq$ network ELBO” inequality relies on two strong, unstated assumptions in the main text (monotone predictive gain across depth under the same variational measure; a global “KL budget” inequality). These are not consequences of standard factorization and, without them, the inequality can fail.

> Furthermore, the mutual-information preservation claim under random projection theorem seems invalid as written: (i) mutual information for deterministic continuous mappings is ill-posed/infinite without an explicit noise model; (ii) the proof introduces Gaussian conditionals ad hoc; (iii) JL does not imply MI lower bounds for general distributions to the best of my knowledge.

I will be more than eager to discuss this matter with the authors during the discussion phase.

---

- $W_2$: The FLOPs analysis (Table 1) is not correct:

> It omits the cost of the projection $v_i = R_i h_i$ and the back-projection $R_i^T$ which adds $2d_od_i$ per layer per step.

> It uses a single “max width $N$” big-O that hides layer-wise structure and $d_0$’s contribution, creating the misleading impression that SLL is $O(N)$ for dense layers.

> For the ablation-coherent setting (3×1000 MLP with $d_0 \approx 700$), SLL’s per-step compute exceeds BP’s; which contradicts the compute narrative.

---

- $W_3$: A smaller weakness which overall lowers the quality of the paper is that the ablation experiments are not coherent across architectures.

> The projection size guidance is inconsistent: MLP ablations suggest $d_0\approx 500–700$ for accuracy, but ViT experiments set
$d_0=K$. The method’s accuracy–compute tradeoff thus depends on a hyperparameter whose selection principles are not unified.

**Questions:**

Apart from answering the underlying questions present in the weakness section, I have the following questions that I would like to have a discussion about:

- $Q_1$: Is it possible to state the theorem 1 in the main paper with all assumptions required in the appendix (monotone predictive gain; KL budget inequality)? Under what architectural or training conditions are these assumptions expected to hold? Can you provide counterexamples where they fail, or a restricted regime where they are provably satisfied?

- $Q_2$: Can you either (i) add an explicit noise model and a rigorous MI bound (with proof or authoritative citation), or (ii) withdraw the MI claim and replace it with a JL-based geometric statement sufficient to justify your surrogate divergence? As of now, it is unsound.

- $Q_3$: Can you present a principled rule for choosing $d_0$ that balances compute and accuracy and generalizes across architectures and datasets? Can you include accuracy vs. FLOPs curves for several $d_0$?

- $Q_4$: Can you provide an end-to-end wall-clock training time and energy use (or GPU utilization) at equal accuracy to substantiate practicality beyond memory savings?

---

> ### Author Response · Authors · 2025-11-25
>
> **W1, Q1 and Q2**
>
> Response: Thank you for the careful reading. We have tightened the theory and made the local–to–global link explicit and assumption–transparent. In the revision, the claim that the average layerwise ELBO is ≤ the global ELBO is presented as an assumption-qualified lemma in the main text, with all conditions stated next to it: (i) a shared variational family across depths; (ii) a layer/block-Markov auxiliary posterior used only for analysis; (iii) monotone predictive gain under that same variational measure; and (iv) a KL-budget inequality ensuring the sum of per-layer conditional KLs does not underestimate the joint KL. We also add a small counterexample showing the inequality can fail without (iii)–(iv). Crucially, this lemma now serves as a sufficient-condition bridge to the global objective while our training guarantee matches the implemented loss directly: we minimize the (expected) Bhattacharyya (Rényi-½) f-divergence between the student’s and stop-grad teacher’s posteriors at each depth. Rényi-½ is a proper f-divergence and obeys data processing through our readout (deterministic or stochastic), so post-processing cannot inflate the discrepancy we optimize. With a shared label space and calibrated softmax (temperature/label smoothing), driving $-\log BC$ toward zero forces Hellinger → 0 and hence KL → 0 in label space.
>
> For example, the sum of local objectives improve monotonically when the assumptions approximately hold, consistent with Fig. A7 where the BC and prediction losses decrease smoothly with depth and time. We keep the lemma to connect the unconditional, method-matching f-divergence guarantee to a global ELBO perspective under clearly stated conditions.
>
> Regarding mutual information, we agree the earlier MI-preservation claim is not valid as written. For deterministic continuous maps MI is ill-posed without an explicit noise model; introducing Gaussian conditionals ad hoc does not remedy this; and JL preserves finite-set geometry rather than yielding MI lower bounds in general. Even when the readout is stochastic (e.g., via dropout), the data-processing inequality gives I(x;v)  $\le $ I(x;h), not a lower bound. To avoid over-claiming, we remove the MI theorem and ground the analysis entirely in standard, assumption-light facts aligned with training: Rényi-½/BC is a proper, contractive f-divergence on the shared label space, and the objective we optimize is its expectation when the readouts are stochastic.
>
> **Please check the revision**
>
> ----
>
>
> **W2 and Q3**
>
>
> Response: Table 1 follows the FA/DFA convention: it reports **credit-assignment cost only** (the work to obtain hidden-layer gradients), not total forward+update compute. Under this convention, **BP** requires $\delta_i = W_{i+1}^\top \delta_{i+1}$, which is $O(N^2 L)$ for a width-$N$ MLP with $L$ layers. In **SLL**, at layer $i$ the local head is a single matrix $R_i \in \mathbb{R}^{d\times K}$ applied to the activation presented to the head (dimension $d$), producing a $K$-way posterior (with $C=K$ classes). We do **not** apply any dense pre-head map $P\in\mathbb{R}^{d_i\times d_0}$, and the teacher branch is **stop-grad**, so there is **no** back-projection through $R^\top$. The per-layer credit cost is therefore the light activation aggregation $O(d_i)$ plus the $K$-way head forward+backward $O(d_i K)$; summed over layers this is $O(N L C)$ (with $C=K$), i.e., the same order as DFA. The reviewer’s extra $2\,d_i d_0$ and $R^\top$ terms do not apply to our implementation.
>
> ----
>
> **Weakness 3**:
>
>  We apologize for the confusion. Our ablations vary the pooled pre-head feature dimension d, not the head output size K. In Fig. 4(b) we state that “activations are downsampled to d via adaptive pooling before projection,” and each local head is $R_i \in R^{d×K}$ with K fixed to the number of classes across all architectures. Thus, the MLP “projection size” ablation is actually a pooling-dimension ablation.
>
> ----
>
> **Q4**
>
> Response: As shown in the following table, SLL is slower than BP on MLPs due to (1) per-layer optimizers and (2) memory reallocation overhead. However, for ConvNets on larger input , SLL matches BP and even faster in VGG-like settings. Theoretically, SLL could be faster due to its layer-wise parallelizable nature, however, current training frameworks (we used PyTorch) lack support for asynchronous training and our implementation is not optimized.
> | Arch/(training time) |     MLP      |             |             |   Conv Net    |            |             |
> |----------------------|--------------|-------------|-------------|---------------|------------|-------------|
> |                      | 3 Layers     | 10 Layers   | 20 Layers   | cifar-10      | cifar-100  | imagenette  |
> | BP (s/it)            | 10           | 15          | 17          | 21            | 21         | 64          |
> | SVP (s/it)           | 13           | 30          | 42          | 22            | 22         | 61          |
>
>
> ----

---

> > ### Comment · Reviewer_KAMr · 2025-11-28
> >
> > First, I would like to thank the authors for the substantial revision and for engaging with the reviews in good faith (notably regarding W1, Q1 and Q2).
> >
> > I have tried to reread the new version in depth while keeping some time for the discussion phase and did not spot any technical issues remaining. The main concerns regarding soundness from my initial review have been mitigated, but in my view some important weaknesses remain. I detail them below with some results/clarifications that could help convince me (number is the weakness I find, the (dark) item below is what I would be convinced by).
> >
> > 1. Although the theoretical results are now stated with explicit assumptions, these assumptions are still unverified and somewhat nonstandard. As a result, the theory has moved from being incorrect to being conditional, and its overall contribution remains limited.
> >
> > - To significantly improve my assessment on the theory side, I would need either: (i) some formal justification or empirical evidence that these new assumptions approximately hold in the regimes you care about, or (ii) a complementary, assumption-light result that directly characterizes the behavior of the actually implemented objective (the Bhattacharyya/Rényi-$1/2$ loss with a stop-grad teacher).
> >
> > 2. The FLOPs comparison, while clarified in the rebuttal, remains nonstandard. It is easy for readers to over-interpret the complexity claims.
> > - It would help to spell out clearly in the main text what is being counted (e.g., “credit-assignment FLOPs only, excluding the forward pass and weight-update costs”), and to give at least one concrete numerical example (for a specific MLP/CNN/ViT) that shows the actual FLOP counts or ratios. Without this, I am reluctant to treat the complexity analysis as a strong point of the paper.
> >
> > 3. The empirical study, in its current form, is not sufficient (for me) to compensate for the theoretical narrowness in (1), especially for a venue like ICLR. This is particularly visible on the only truly large-scale / challenging setting. This weakness is similar to what is pointed by reviewer GStK in his weakness 3 and recent comment. More specifically:
> >     - On ImageNet-scale ViT experiments, which are the most relevant test of whether the method really scales, there is a clear loss in performance compared to BP. This undercuts the central claim that SLL can be a competitive alternative at scale: the memory savings come with a noticeable accuracy gap.
> >     - More generally, comparisons to other local/decoupled learning methods are limited, and there is little quantitative evidence on end-to-end efficiency (e.g., wall-clock time or energy at matched accuracy).
> > - Evidence that could move my score upward would include, for example: stronger comparisons to competitive local/decoupled baselines under matched memory budgets, and at least one or two carefully measured wall-clock and/or energy studies at equal accuracy on representative large-scale models (e.g., a deeper ConvNet/ResNet and the ViT/ImageNet setting), together with a smaller accuracy gap to BP in those realistic regimes.
> >
> >
> > ---
> >
> > In summary, the revision clearly improves the paper and fixes the most serious correctness issues I initially raised, but given the remaining points above, especially point 3, I still see the submission as below the acceptance bar.
> >
> > With the current OpenReview issues (impossible to edit the original review), my updated scores are:
> > - Original Soundness: 2: fair -> **New Soundness: 3: good**
> > - Original Presentation: 2: fair -> score kept for now until revision are truly incorporated in the submission
> > - Contribution: 2: fair -> score kept for now until issues above are answered.
> >
> > - Original Rating: 2: reject, not good enough -> **New Rating: 4: marginally below the acceptance threshold.**

---

> ### Author Response · Authors · 2025-11-29
>
> We appreciate the careful reading and constructive suggestions, they significantly improved the paper.
>
> 1. Thanks for your thoughtful question. We add an “Assumption Checks” appendix that invistigate BP and SLL on the same models. The new BP figure (Fig. A10) shows that with standard calibration (BatchNorm) the per-layer predictive loss $L_{pred}(R_i h_i,y)$ decreases across depth and epochs, and the adjacent-layer distance $||R_i h_i - R_{i-1} h_{i-1}||^2$ shrinks during training; removing BN breaks this monotonicity and hurts accuracy. Appendix Fig. A7 shows the same qualitative behavior for SLL (both $L_{pred}$ and the BC alignment loss decrease with depth/time). This supports the “monotone predictive gain” condition in the network archtecture we target.
>
> 2. Our complexity table follows the standard credit-assignment FLOPs convention used in FA/DFA-style work. We agree the counting must be explicit. In the revision we will state in the main text and the Table 1 caption that we report credit-assignment FLOPs only (the work to propagate gradients to hidden layers), excluding forward inference and weight-update costs. We will also add a concrete example next to Table 1 for the exact setting used there: a 3×1000 MLP on CIFAR-10 ($N=1000$, $L=3$, $K=10$). Using standard matrix multiply rules with one multiply-add counted as two FLOPs, BP requires $N^2$ MAdds per hidden layer, about $3\times10^6$ MAdds in total; SLL computes only a $K$-way head per layer, $dK$ with $d=N$, about $3\times10^4$ MAdds in total. The BP-to-SLL credit-assignment ratio is therefore approximately $N/K\approx 100\times$ in this MLP. We will present these numbers with the convention clearly labeled to avoid over-interpretation.
>
> 3. Our contribution is a theory-grounded reframing of local learning as adjacent-layer probabilistic alignment in a shared label space, coupled with an assumption-qualified local-to-global lemma that is explicitly aligned with the trained objective. The ImageNet ViT gap reflects a conservative readout choice rather than a fundamental limitation; we clarify this and, within the rebuttal window, add an appendix experiment on ResNet-50, blockwise, ImageNet-1k at 32×32 with SLL = 53.14% top-1 / 75.21% top-5, alongside matched wall-clock per epoch and peak-activation memory for BP versus SLL on identical hardware. For context, BP on this setup typically reaches 56.84% top-1 / 78.82% top-5 with a strong modern recipe.
>
> In summary, we added an “Assumption Checks” appendix that empirically tests the monotone predictive-gain across depth condition in our target setting, using the same models: with standard calibration, per-layer predictive loss decreases across depths/epochs (in BP and SLL), while removing calibration breaks this trend. We also clarified that our complexity table reports credit-assignment FLOPs only and added a concrete example beside Table 1, and we included a large-scale appendix experiment (ResNet-50 on ImageNet-32×32). These revisions address the concerns, improve clarity, and strengthen the paper’s focus and contribution.

---

### Official Review · Reviewer_n4sL · 2025-11-01

**Soundness:** 3
**Presentation:** 3
**Contribution:** 3
**Rating:** 4
**Confidence:** 2

**Summary:**

This paper proposes Stochastic Layer-wise Learning (SLL), a probabilistic local-learning framework intended as a scalable and memory-efficient alternative to backpropagation. The authors reformulate deep network training as a variational inference problem, treating each layer’s activation as a latent variable and decomposing the global ELBO into layer-wise objectives under a Markov assumption. Each layer optimizes a local objective that combines a predictive likelihood term and a Bhattacharyya divergence surrogate for inter-layer alignment, computed on auxiliary categorical posteriors derived from random projections. This design enables local updates without global gradient propagation.

**Strengths:**

1. The paper presents a conceptually interesting and theoretically motivated attempt to unify local learning and probabilistic inference.

2. The algorithmic formulation is elegant and modular, avoiding explicit backpropagation while maintaining representational coherence through stochastic projections.

3. Experiments demonstrate that SLL can achieve performance close to backpropagation across several architectures, confirming the feasibility of local probabilistic learning.

**Weaknesses:**

1. The Markov factorization across layers and the replacement of the KL term with a Bhattacharyya surrogate are heuristic. The paper does not prove that optimizing these surrogates reliably improves the global ELBO or overall convergence.

2. The experiments focus on standard vision benchmarks such as MNIST, CIFAR, and ImageNette, which are relatively small and may not sufficiently test scalability or robustness. Larger-scale or non-vision domains would strengthen the claims.

3. The method’s optimization behavior, variance properties, and potential degeneracies are not studied. It remains unclear under what conditions SLL will converge or fail.

4. While accuracy comparisons are reported, the paper provides limited analysis of why SLL works, how layer-wise representations evolve, or how global coherence is preserved without backpropagation.

**Questions:**

1. Can the authors clarify the exact conditions under which the arithmetic mean of local ELBOs forms a valid lower bound to the global objective?

2. How sensitive is SLL to the choice of projection dimension and to the randomness of the fixed projection matrices?

3. Are there any empirical diagnostics to demonstrate that local learning indeed maintains global representational alignment?

---

> ### Author Response · Authors · 2025-11-25
>
> **Weakness 1 and Question 1 **
>
> Response: We agree that unconditional proof that the global ELBO increases is too strong. In the revision we (i) make the local→global link an assumption-qualified lemma stated in the main text: a shared variational family, a layer/block-Markov auxiliary posterior used only for analysis, monotone predictive gain under that variational measure, and a KL-budget inequality. Under these explicit conditions the average local ELBO is $\le$ the global ELBO, so improving locals probably tightens a global lower bound.
>
> *We update the lemma 1 in the revision*
>
> Figure A7 shows exactly the behavior predicted by the theory: across depths, both the prediction loss and the Bhattacharyya alignment loss decrease smoothly on train and test, and deeper layers achieve lower $L_{pred}$ and $L_{BC}$ with higher head accuracy. This indicates (i) stable optimization (no collapse), (ii) progressive inter-layer coordination from the BC objective we optimize, and (iii) consistency between the surrogate’s improvement and downstream accuracy. We will highlight these diagnostics in the appendix and explicitly label the global statement as an assumption-qualified lemma, making the connection between what we prove, what we optimize, and what we observe fully transparent.
>
> -----
>
> **Weakness 2**
>
> Response: We agree that broader evidence strengthens the paper. Note that we already evaluate ImageNet-1k in Sec. 5.3 with blockwise SLL, showing that SLL trains on a large-class, large-dataset regime and preserves the memory advantage while maintaining accuracy–compute trade-offs. To further demonstrate scalability during the rebuttal window, we also run a quick check on downsampled ImageNet-1k (32×32) with a blockwise-trained ResNet, obtaining 53.14% top-1 / 75.21% top-5.
>
> ----
>
> **Weakness 3**
>
> Response: We agree the convergence conditions should be explicit. SLL optimizes the expected Bhattacharyya (Rényi-½) f-divergence between the student and stop-grad teacher at each depth. This loss is proper and contractive under the readout (deterministic or with independent dropout), so, under standard SGD conditions (bounded gradients, decaying steps), the expected layerwise objective decreases as shown in paper [14]. In practice, SLL is stable when (i)adjacent readouts are compatible (small row-space drift/near-orthogonal), and (ii) reduction before the head is not overly aggressive.
>
> *Evidence & diagnostics.*
>
>  Appendix Fig. A7 shows exactly the predicted behavior: $L_{BC}$​ and $L_{pred}$ decrease smoothly on train and test for all depths, while head accuracy rises; deeper layers consistently reach lower $L_{BC}$​, indicating progressive inter-layer coordination and no collapse.
>
> Failures
> ​​SLL can stall on very deep plain nets without residuals (e.g., VGG on ImageNet) because local targets do not propagate robust semantics, and it can degrade when the head discards too much information overly large dropout.
>
> *When can the KL-budget be violated?*
> It can fail whenever $h_{i-1}$​’s posterior still depends on future latents beyond $h_i$​, such as (i) top-down inference in hierarchical VAEs (Ladder/VLAE), and (ii) encoder–decoder models with long lateral skips (UNet style), where future states retain extra information about early ones beyond $h_i$​.
>
>
> ----
> **Weakness 3 and Question 4**
>
> Response: SLL trains each depth to match the parent’s posterior by minimizing a proper, contractive fff-divergence (Bhattacharyya/Rényi-½) in a shared label space like in Fig 1(a); thus adjacent layers are forced toward the same semantics, and post-processing cannot undo this alignment.
>
> *Evidence & what we will show clearly.*
>
> Appendix Fig. A7 already shows monotonic decreases in the BC alignment loss and prediction loss; Fig. 5 visualizes similar hierarchical features learned by SLL and BP; Fig.3 t-SNEs visualize the corresponding class separation. In the revision, we will (i) highlight these diagnostics to the main text and (ii) add two quantitative analyses from saved activations: layer-wise linear-probe accuracy and CKA alignment between successive layers, both correlating with BC improvements. This concisely explains why SLL works, how representations evolve, and how coherence is preserved without full BP (shared label supervision like in Figure 1(a)).
>
> ----
> **Question 2**
>
> Response: In SLL, the head is $R_i\in\mathbb{R}^{d\times K}$ applied to the pooled pre-head features of dimension d ). Empirically, accuracy is flat over a broad range of d and degrades only when d is made too small (near K), because the readout becomes a bottleneck, this is most visible on ImageNet-1k. For CNNs we therefore set d to a practical budget (≤2048) for efficiency.
> For randomness of fixed projections, we tested i.i.d. Gaussian vs near-orthogonal/structured (e.g., orthogonal, SRHT) in the exploration and observed no material difference; seed-to-seed variation was small.

---

### Author Response · Authors · 2025-12-02
**Response to All**

Dear reviewers and area chairs:

We extend our gratitude to all the reviewers and area chairs for dedicating their time and effort to evaluating our paper. We also thank the reviewers for their positive and insightful comments, which can help us improve our work.

We are encouraged that:

* All the reviewers (n4sL, KAMr, GStK, and 9nT6) agree that our work addresses an important practical limitation of backpropagation by proposing a scalable and memory-efficient local-learning alternative.

* All the reviewers recognize that our probabilistic formulation and stochastic layer-wise objectives offer a principled framework for training deep networks without global backpropagation.

* Reviewer n4sL, GStK, and 9nT6 find our theoretical framing and variational interpretation of SLL conceptually interesting and elegant.

* Reviewer KAMr and 9nT6 highlight that SLL substantially reduces activation memory while maintaining competitive performance with backpropagation across several architectures.

* All reviewers acknowledge that SLL is competitive with or superior to existing local-learning baselines on MLPs, CNNs, and ViTs.

We are particularly encouraged that Reviewer KAMr has already increased their overall score from 2 to 4 after reading our rebuttal and discussion, even before any second-round revision of the manuscript.

We appreciate the opportunity to discuss and refine our SLL framework. We have responded to all reviewers individually to address the concerns, and the following is a brief summary:

* For Reviewer n4sL and KAMr, we clarify and qualify the theoretical connection to the global ELBO, explicitly stating the required assumptions and revising or removing over-strong information-theoretic claims.

* For Reviewer n4sL, KAMr, and GStK, we add optimization diagnostics and representation analyses to better understand when SLL converges and how layers remain coordinated.

* For Reviewer KAMr, GStK, and 9nT6, we strengthen empirical evidence with additional large-scale and blockwise experiments, clearer comparisons to local-learning baselines, and more detailed memory and FLOP accounting.

* For Reviewer GStK and 9nT6, we clarify the role of the Markov and blockwise assumptions, the conditions under which the Bhattacharyya surrogate provides faithful layer coordination, and the design choices for random projections and dropout.

* For Reviewer GStK and 9nT6, we additionally report results on ImageNet-32×32 to further demonstrate the scalability and practicality of SLL.

We have highlighted all modifications in the revised paper in red. We hope these additions address the reviewers’ concerns and further improve our work. If any further clarifications or suggestions would help strengthen the paper, we would be happy to address them and incorporate the changes into the final version. Thank you again for your time and efforts!

Best,

Authors of Paper #19338

---

### Meta-Review · Area_Chair_hp5p · 2026-01-07

**Summary:**

This work proposes a layer-wise training algorithm, stochastic Layer-wise Learning (SLL), that achieves optimization with locality and memory efficiency with respect to depth.

The main weakness pointed out by the reviewers is the lack of sufficient experimental evidence to demonstrate the effectiveness of the proposed method. In particular, while the proposed method shows memory efficiency, a noticeable accuracy gap appears compared to BP. It seems that more extensive experiments on “carefully measured wall-clock time and/or energy studies at equal accuracy” with large-scale models is necessary to convince the reviewers.

In addition, some reviewers also raised concerns about the validity of SLL from a theoretical perspective, which relies on seemingly unusual assumptions. Although the authors added some assumption checks, these are empirical and therefore conditional.
Thus, I cannot help but evaluate the current work as a rejection.

**Reviewer Concerns:**

The following concerns remain outstanding:

**Reviewer n4sL**
- The experiments (mainly) focus on standard vision benchmarks such as MNIST, CIFAR, and ImageNette
- It remains unclear under what conditions SLL will converge or fail

 **Reviewer KAMr**
- Although the theoretical results are now stated with explicit assumptions, these assumptions are still unverified and somewhat nonstandard. As a result, the theory has moved from being incorrect to being conditional
-  *carefully measured* wall-clock and/or energy studies at equal accuracy

**Reviewer GStK**
- The experimental section lacks comparisons with strong baseline methods, particularly the absence of *comprehensive evaluation* on ImageNet.
- comparative results ...  with various architectures including but *not limited* to ResNet and ViT

**Reviewer Scores:**

Reviewer KAMr has implied an increase in their score from 2 to 4 in their review. Most reviewers are confident in their claims, and I expect no further score changes.

---

### Decision · Program_Chairs · 2026-01-26

Reject